# TNFα drives pulmonary arterial hypertension by suppressing the BMP type-II receptor and altering NOTCH signalling

Liam A. Hurst[1,*], Benjamin J. Dunmore[1,*], Lu Long[1], Alexi Crosby[1], Rafia Al-Lamki[1], John Deighton[1], Mark Southwood[2], Xudong Yang[1], Marko Z. Nikolic[1], Blanca Herrera[3], Gareth J. Inman[4], John R. Bradley[1], Amer A. Rana[1,**], Paul D. Upton[1,**] & Nicholas W. Morrell[1,**]

Heterozygous germ-line mutations in the bone morphogenetic protein type-II receptor (BMPR-II) gene underlie heritable pulmonary arterial hypertension (HPAH). Although inflammation promotes PAH, the mechanisms by which inflammation and BMPR-II dysfunction conspire to cause disease remain unknown. Here we identify that tumour necrosis factor-α (TNFα) selectively reduces BMPR-II transcription and mediates post-translational BMPR-II cleavage via the sheddases, ADAM10 and ADAM17 in pulmonary artery smooth muscle cells (PASMCs). TNFα-mediated suppression of BMPR-II subverts BMP signalling, leading to BMP6-mediated PASMC proliferation via preferential activation of an ALK2/ACTR-IIA signalling axis. Furthermore, TNFα, via SRC family kinases, increases pro-proliferative NOTCH2 signalling in HPAH PASMCs with reduced BMPR-II expression. We confirm this signalling switch in rodent models of PAH and demonstrate that anti-TNFα immunotherapy reverses disease progression, restoring normal BMP/NOTCH signalling. Collectively, these findings identify mechanisms by which BMP and TNFα signalling contribute to disease, and suggest a tractable approach for therapeutic intervention in PAH.

[1] Division of Respiratory Medicine, Department of Medicine, Box 157, Level 5, University of Cambridge School of Clinical Medicine, Addenbrooke's and Papworth Hospitals, Hills Road, Cambridge CB2 0QQ, UK. [2] Department of Pathology, Papworth Hospital, Papworth Everard, Cambridge CB23 8RE, UK. [3] Department Bioquímica y Biología Molecular II, Facultad de Farmacia, Universidad Complutense. Instituto de Investigación Sanitaria del Hospital Clínico San Carlos (IdISSC), Calle Del Prof Martin Lagos, Madrid 28040, Spain. [4] Division of Cancer Research, Jacqui Wood Cancer Centre, School of Medicine, University of Dundee, Ninewells Hospital And Medical School, Dundee DD1 9SY, UK. * These authors contributed equally to this work. ** These authors jointly supervised this work. Correspondence and requests for materials should be addressed to P.D.U. (email: pdu21@medschl.cam.ac.uk) or to N.W.M. (email: nwm23@cam.ac.uk).

Pulmonary arterial hypertension (PAH) is a progressive disease defined by elevated pulmonary arterial pressure, often causing death from right heart failure. The pathology is characterized by increased muscularization and obliteration of small pulmonary arteries[1]. Heterozygous germ-line mutations in the *BMPR2* gene, encoding the bone morphogenetic protein type II receptor (BMPR-II), underlie ~70% of heritable (HPAH) and 20% of idiopathic (IPAH) cases[2–4]. Most *BMPR2* mutations cause haploinsufficiency and importantly, pulmonary vascular BMPR-II levels are reduced in non-genetic forms of PAH in animals and humans[5–7].

Despite *BMPR2* mutations being the commonest genetic cause for PAH, the penetrance of mutations in carriers is only 20–30%, suggesting that additional factors are required for disease initiation and progression. Inflammation is strongly implicated as a trigger for disease[4,8,9] and promotes the development of PAH in *Bmpr2*[+/−] mice[10]. Accordingly, PAH patients exhibit heightened circulating levels of inflammatory cytokines, including tumour necrosis factor-α (TNFα), that correlate with poor survival[11,12]. Of note, transgenic mice overexpressing TNFα in the lung develop spontaneous PAH[13] and lung TNFα expression is elevated in rats or dogs with pulmonary hypertension induced with monocrotaline (MCT-PAH) or a high flow left-to-right shunt[14–16]. Etanercept, a soluble TNF-receptor II dimer, prevents and reverses MCT-PAH in rats[17] and reverses PAH in endotoxemic pigs[18]. Moreover, TNFα suppresses BMPR-II levels in aortic endothelial cells[19]. Although these observations suggest TNFα activity and aberrant BMPR-II signalling interact in PAH, a direct molecular mechanism remains elusive.

BMPR-II forms heteromeric cell surface receptor complexes with activin-like kinase (ALK) type I receptors[20], mediating BMP2, BMP4 and BMP6 signalling with ALK3 in pulmonary artery smooth muscle cells (PASMCs)[21], or mediating endothelial BMP9/10 responses with ALK1 (refs 22,23). The activated receptors phosphorylate the canonical Smad1/5/8 proteins[20] that promote the transcription of genes including the Inhibitor of DNA-binding (ID) gene family and NOTCH pathways[24,25]. BMPs can also signal independently of Smads, through mitogen activated protein kinases (MAPKs), and proto-oncogene protein tyrosine kinase c-SRC (c-SRC) phosphorylation[26,27]. We have previously shown reduced BMP4-dependent Smad1/5/8 signalling and transcriptional responses in PASMCs from PAH patients, especially those harboring *BMPR2* mutations[28,29]. In contrast to reduced BMP4 signalling, deletion of both *Bmpr2* alleles in mouse PASMCs, or small interfering RNA (siRNA)-mediated knockdown of BMPR-II protein in human PASMCs, enhances BMP6 and BMP7-mediated Smad signalling via recruitment of ACTR-IIA and ALK2 (refs 21,30,31). Since TNFα reduces endothelial *BMPR2* expression[19], we hypothesized that TNFα may critically reduce *BMPR2* expression in vascular cells harboring *BMPR2* mutations and switch BMP signalling to recruit ACTR-IIA and ALK2, with potentially pathological consequences.

Here we demonstrate that TNFα, a key inflammatory mediator, reduces *BMPR2* expression in vascular cells and promotes ADAM10/17-dependent BMPR-II cleavage in PASMCs, releasing the soluble ectodomain which acts a ligand trap. Furthermore, we identify the mechanism by which TNFα, against a background of *BMPR2* genetic loss-of-function, promotes the development of PAH by driving inappropriate PASMC proliferation through c-SRC family members and dysregulated NOTCH2/3 signalling. Moreover, therapeutic etanercept administration reversed PAH progression in the rat Sugen-hypoxia model and redressed the NOTCH imbalances. This provides a rationale for the development of anti-TNF strategies for the treatment of PAH.

## Results

**TNFα reduces BMPR-II expression in vitro and in vivo.** Several cytokines, including TNFα, IL-1β, IL-6 and IL-8 are implicated in the pathogenesis of PAH[9–13,16]. Of these, only TNFα selectively reduced *BMPR2* mRNA and BMPR-II protein in distal PASMCs (dPASMCs) and pulmonary arterial endothelial cells (PAECs) (Fig. 1a–d and Supplementary Fig. 1a–d), via NF-κB p65 (*RELA*; Supplementary Fig. 1e). Furthermore, immunofluorescent staining demonstrated local vascular expression of TNFα in both human IPAH and HPAH that was absent in unaffected controls (Fig. 1e).

**TNFα induces BMPR-II extracellular domain shedding in SMCs.** Unexpectedly, in dPASMCs (Fig. 1c), but not PAECs (Fig. 1d), the TNFα-mediated reduction of full length BMPR-II levels (140–150 kDa) was associated with accumulation of an intracellular 60 kDa product (BMPR-II-ICP), confirmed as a BMPR-II fragment using siRNA (si*BMPR2*; Supplementary Fig. 2a). Furthermore, TNFα also promoted the production of this 60 kDa band and reduction of full length BMPR-II in rat and mouse PASMCs, human proximal PASMCs (pPASMCs), and human aortic smooth muscle cells (Supplementary Fig. 2b,c).

To confirm these observations *in vivo*, we examined BMPR-II expression in *SP-C/Tnf* mice[13], which overexpress mouse TNFα in the lung and developed PAH by 8 weeks of age (Fig. 1f and Supplementary Fig. 3a–c). *SP-C/Tnf* mice exhibited reduced *Bmpr2* mRNA and BMPR-II protein and accumulation of BMPR-II-ICP in lung, but not liver (Fig. 1g and Supplementary Fig. 3d,e).

The presence of the BMPR-II-ICP in PASMCs and *SP-C/Tnf* lung suggested TNFα-dependent cleavage of BMPR-II. We confirmed this through immunoprecipitation of a myc-tagged BMPR-II ectodomain from conditioned media from TNFα-treated dPASMCs (Supplementary Fig. 4a). Furthermore, ELISA of conditioned media from TNFα-treated PASMCs revealed enhanced endogenous soluble BMPR-II (sBMPR-II) generation (Supplementary Fig. 4b). Since BMPR-II cleavage has not been reported previously, we determined the proteolytic mechanism of TNFα-mediated cleavage of BMPR-II in PASMCs. Previous studies demonstrated that matrix metalloproteinase-14 (MMP-14) cleaves the TGFβ co-receptors, endoglin and betaglycan[32,33] and A Disintegrin and Metalloprotease-17 (ADAM17) mediates TGFβ type-I receptor ectodomain shedding[34]. Accordingly, a pan-MMP/ADAM inhibitor, batimastat (BB94), inhibited the TNFα-dependent BMPR-II cleavage and sBMPR-II generation (Supplementary Fig. 4c,d). Transcriptional analysis of candidate metalloproteinases revealed that TNFα induced *ADAM10* and *ADAM17* in dPASMCs, but not PAECs (Supplementary Fig. 5a,b) and ADAM10 and ADAM17 were increased in *SP-C/Tnf* mouse lung homogenates (Supplementary Fig. 5c). Since ADAM10 and ADAM17 levels do not reflect altered activity, we examined directly whether either ADAM was responsible for the BMPR-II cleavage[35]. Interestingly, only dual ADAM10/17 inhibition (Supplementary Fig. 5d,e) or combined siRNAs (Fig. 1h) prevented BMPR-II cleavage and sBMPR-II generation from PASMCs, confirming that both ADAM10 and ADAM17 cleave BMPR-II with dual redundancy.

*In silico* analysis of published ADAM10/17 cleavage sites suggested selectivity for alanine-valine (Ala-Val) junctions and we identified four valines within the transmembrane domain potentially permitting sBMPR-II generation (Supplementary Fig. 6a). Mutagenesis of each valine residue demonstrated that the V163A mutation completely prevented BMPR-II cleavage (Supplementary Fig. 6b) and sBMPR-II generation (Fig. 1i).

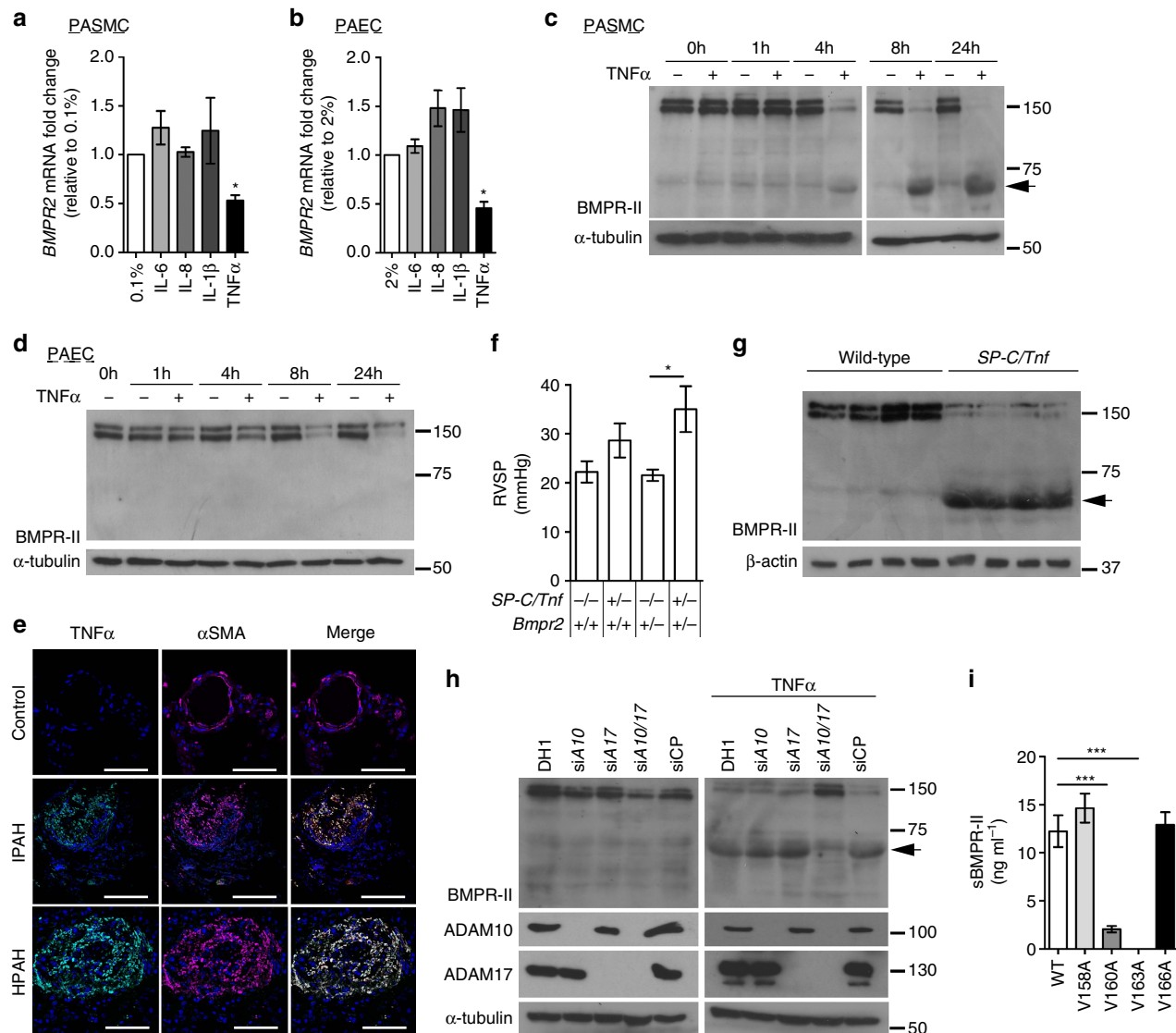

**Figure 1 | TNFα reduces BMPR-II expression and promotes BMPR-II cleavage in PASMCs.** (**a**,**b**) *BMPR2* mRNA expression, normalized to *ACTB*, in human dPASMCs (**a**) and PAECs (**b**) treated with IL-1β (1 ng ml$^{-1}$), IL-6 (25 ng ml$^{-1}$), IL-8 (25 ng ml$^{-1}$) or TNFα (1 ng ml$^{-1}$) for 24 h ($n=3$; Student's *t*-test). (**c**,**d**) Representative immunoblots of BMPR-II expression in human dPASMCs (**c**) and PAECs (**d**) treated with TNFα (1 ng ml$^{-1}$) for 1, 4, 8 or 24 h. Reprobed for α-tubulin to ensure equal loading. The data shown are representative of three experiments. (**e**) Representative confocal images of immunohistochemical staining for TNFα (turquoise) and αSMA (magenta) in lung sections from control, idiopathic and heritable PAH subjects. Nuclei were counterstained with DAPI (blue). Scale bars, 100 μm. (**f**) Assessment of right ventricular systolic pressure (RVSP) from *Bmpr2*$^{+/+}$, *SP-C/Tnf/Bmpr2*$^{+/+}$, *Bmpr2*$^{+/-}$ and *SP-C/Tnf/Bmpr2*$^{+/-}$ ($n=4$ per group) mice. (**g**) Representative immunoblots of BMPR-II expression in lungs isolated from 8 week old *Bmpr2*$^{+/+}$ and *SP-C/Tnf/Bmpr2*$^{+/+}$ transgenic mice. Reprobed for β-actin to ensure equal loading ($n=4$). (**h**) Representative immunoblots of BMPR-II, ADAM10 and ADAM17 in human dPASMCs transfected with DharmaFECT1 alone (DH1), si*ADAM10*, si*ADAM17*, combined si*ADAM10* + si*ADAM17* (siADAM10/17) or non-targeting siRNA control (siCP) with or without TNFα (1 ng ml$^{-1}$) treatment for 24 h. Reprobed for α-tubulin to ensure equal loading. The data shown are representative of three experiments. (**i**) ELISA measurement of soluble BMPR-II in conditioned media from human dPASMCs transfected with wild-type and mutant 5′-myc-tagged BMPR-II constructs and treated with TNFα (1 ng ml$^{-1}$) for 24 h ($n=3$). One-way analysis of variance with *post hoc* Tukey's for multiple comparisons used in **f** and **i**. *$P\leq 0.05$, ***$P\leq 0.001$. Error bars represent mean ± s.e.m. Lower molecular mass BMPR-II is indicated by an arrow.

Many soluble receptor ectodomains function as ligand traps. Accordingly, commercially sourced recombinant BMPR-II ECD (Supplementary Fig. 6c), or conditioned media from TNFα-treated dPASMCs overexpressing wild-type BMPR-II (Supplementary Fig. 6d), inhibited BMP2 and BMP4 signalling, whereas media from the cleavage-resistant V163A mutant did not inhibit these responses (Supplementary Fig. 6d). Of note, BMPR-II ECD neutralized the anti-proliferative effects of BMP2 and BMP4 (Supplementary Fig. 6e), in a similar manner to treatment with TNFα (Supplementary Fig. 6f).

**TNFα alters BMP signalling and promotes PASMC proliferation.** Having demonstrated that TNFα suppresses *BMPR2* expression *in vitro* and *in vivo*, we questioned the impact on BMP signalling in the context of an existing *BMPR2* mutation. We confirmed that the reduced expression of *BMPR2* in dPASMCs from patients with heritable PAH (HPAH) is further reduced by TNFα (Fig. 2a and Supplementary Fig. 7a). Previous studies reported that BMPR-II loss in PASMCs reduces BMP2 and BMP4 signalling, but reveals gain-of-signalling to BMP6 or BMP7 via ALK2 and ACTR-IIA[21,30,31]. This enhanced BMP7 response

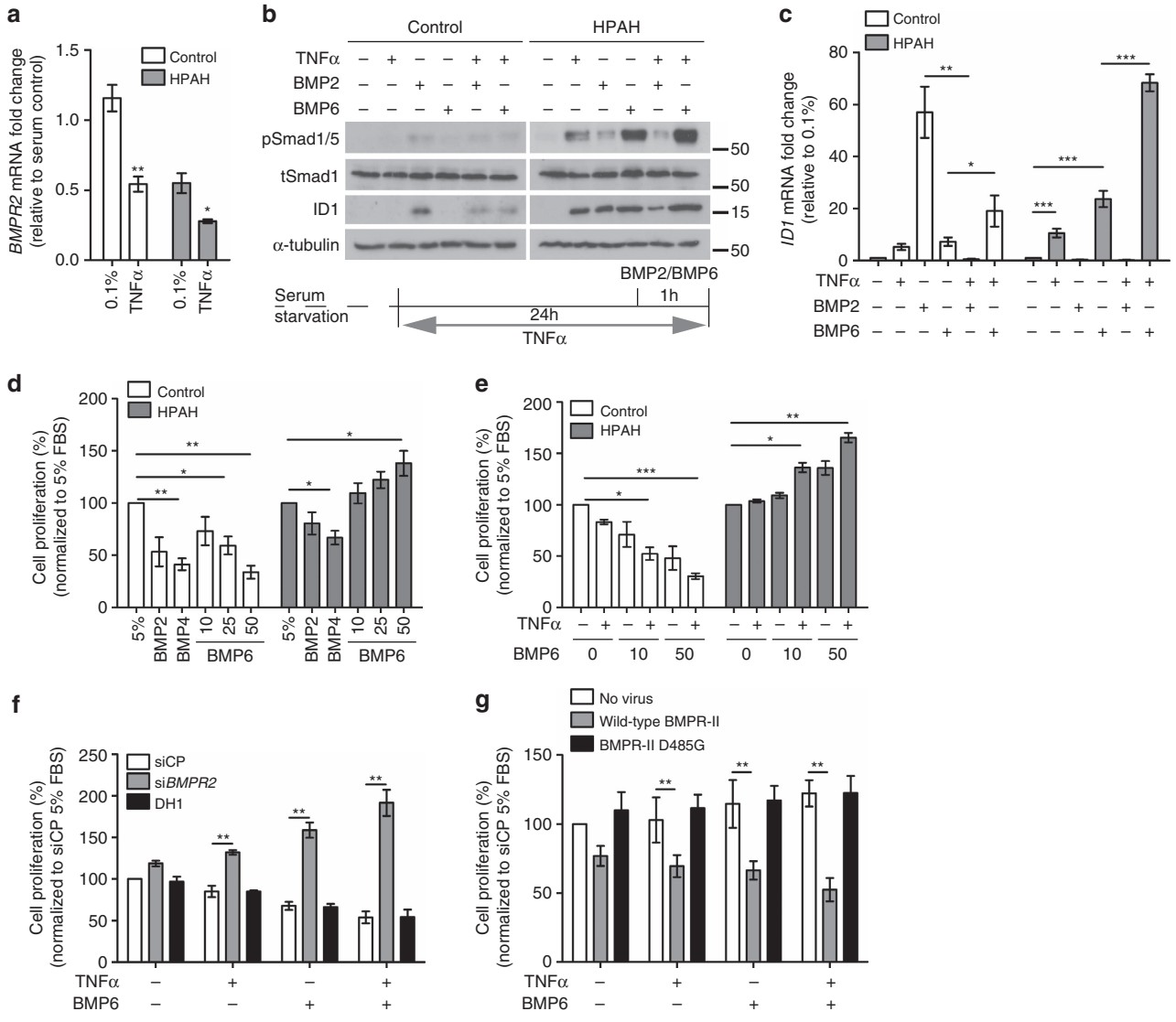

**Figure 2 | TNFα alters BMP2 and BMP6 signalling dynamics and PASMC proliferation.** (**a**) *BMPR2* mRNA expression, normalized to *ACTB*, in human dPASMCs from disease-free controls and HPAH patients stimulated with TNFα (1 ng ml$^{-1}$) for 24 h (n = 3; Student's *t*-test). (**b,c**) Pre-treatment of human dPASMCs from disease-free controls and HPAH patients with or without TNFα (1 ng ml$^{-1}$) for 23 h before 1 h stimulation with BMP2 (10 ng ml$^{-1}$) or BMP6 (10 ng ml$^{-1}$). (**b**) Representative immunoblots of phospho-Smad1/5, total Smad1 and ID1 expression. Reprobed for α-tubulin to ensure equal loading. The data shown are representative of three control and HPAH cell lines. (**c**) *ID1* mRNA expression, normalized to *ACTB* (n = 3). (**d**) Proliferation of human dPASMCs from disease-free controls and HPAH patients after 6 days treatment. Cells treated every 48 h as indicated with BMP2 (10 ng ml$^{-1}$), BMP4 (10 ng ml$^{-1}$) or BMP6 (10, 25 or 50 ng ml$^{-1}$; n = 3). (**e**) Proliferation of human dPASMCs from disease-free controls and HPAH patients after 6 days treatment. Cells treated every 48 h as indicated with TNFα (1 ng ml$^{-1}$) and/or BMP6 (10 or 50 ng ml$^{-1}$) (n = 3 control and HPAH cell lines). (**f**) Proliferation of human control dPASMCs on day 6 after transfection with DharmaFECT1 alone (DH1), si*BMPR2* or non-targeting siRNA control (siCP) and treatment every 48 h with TNFα (1 ng ml$^{-1}$) and/or BMP6 (50 ng ml$^{-1}$) as indicated (n = 3). (**g**) Proliferation of human HPAH dPASMCs on day 6 following transduction with adenovirus expressing full-length wild-type and kinase-dead D485G mutant BMPR-II cDNA and treatment every 48 h with TNFα (1 ng ml$^{-1}$) and/or BMP6 (50 ng ml$^{-1}$) as indicated (n = 3). One-way analysis of variance with *post hoc* Tukey's for multiple comparisons used in (**c-g**). *P ≤ 0.05, **P ≤ 0.01, ***P ≤ 0.001. Error bars represent mean ± s.e.m.

was reported to be transient[31] and here we confirmed that siRNA-mediated loss of BMPR-II in control PASMCs led to an enhanced BMP6-dependent Smad1/5 phosphorylation at 1 h that was not observed at 4 or 24 h (Supplementary Fig. 7b,c). As expected, TNFα inhibited BMP2-dependent Smad1/5 phosphorylation and *ID1* transcription in control dPASMCs, but augmented BMP6 signalling, particularly in HPAH PASMCs (Fig. 2b,c and Supplementary Fig. 7d,e). Functionally, BMP2, BMP4 and BMP6 inhibited control dPASMC proliferation whereas BMP6 promoted HPAH dPASMC proliferation

and TNFα enhanced the BMP6 responses (Fig. 2d,e). The pivotal role of BMPR-II levels in this TNFα/BMP6 response was demonstrated by the switching from anti-proliferative to pro-proliferative responses following si*BMPR2* in control dPASMCs (Fig. 2f), and restoration of the anti-proliferative response to BMP6 following overexpression of wild-type BMPR-II in HPAH dPASMCs (Fig. 2g).

In pulmonary vascular cells, BMP2 and BMP6 are relatively highly expressed (Supplementary Fig. 8a,b). Interestingly, TNFα repressed *BMP2*, but consistently induced *BMP6* expression in

dPASMCs and PAECs (Supplementary Fig. 8c,d), while other BMP ligands were unaltered (Supplementary Fig. 8e). The BMP2 and BMP6 transcriptional responses were mediated through NF-κB p65 (*RELA*) in dPASMCs and PAECS (Supplementary Fig. 8f,g). Furthermore, BMP6 induction by TNFα was greater in HPAH dPASMCs than control cells whereas BMP2 expression was repressed equally (Supplementary Fig. 8h,i). Immunoneutralization using anti-BMP6 or LDN193189, an inhibitor of ALK2/3/6 (ref. 36) and ACTR-IIA[37], in C2C12-BRE cells reduced the response to TNFα (Supplementary Fig. 9a,b). Also, LDN193189 inhibited the *ID1* response to TNFα in dPASMCs without affecting the *IL8* response, confirming the *ID1* response is indirectly via BMP receptors and not via canonical NF-κB signalling (Supplementary Fig. 9c,d). Collectively, these data demonstrate that TNFα reduces *BMPR2* and *BMP2* expression, but increases *BMP6* expression in pulmonary vascular cells.

To confirm the *in vivo* relevance of the above, we crossed the *SP-C/Tnf* mouse with a *Bmpr2*$^{+/-}$ mouse that does not develop significant PAH at baseline[38]. The highest mean RVSP was observed in the *SP-C/Tnf/Bmpr2*$^{+/-}$ mice, although not significantly different from the mean RVSP measured in *SPC/Tnf/Bmpr2*$^{+/+}$ mice (Fig. 1f). Right ventricular hypertrophy was elevated to similar extents in both genotypes (Supplementary Fig. 3a), probably due to the high levels of TNFα expression in this model. However, we observed a greater repression of *Bmpr2* and enhancement of *Bmp6* expression in *SP-C/Tnf/Bmpr2*$^{+/-}$ mice compared with *SPC/Tnf/Bmpr2*$^{+/+}$ mice (Supplementary Fig. 3d,f), consistent with our observations that TNFα may dysregulate BMP signalling on the background of *Bmpr2* haploinsufficiency. Lung TNFα overexpression also promoted pulmonary arteriolar muscularization (Supplementary Fig. 3b,c) and repressed *Bmp2* (Supplementary Fig. 3g), albeit to similar extents in *SP-C/Tnf/Bmpr2*$^{+/+}$ and *SP-C/Tnf/Bmpr2*$^{+/-}$ mice.

**BMP6 drives HPAH PASMC proliferation via ALK2 and ACTR-IIA**. We next determined the receptors utilized by BMP6 to promote PASMC proliferation. Both LDN193189 (Fig. 3a,b) and *ALK2* siRNA (Fig. 3c,d) abolished the anti-proliferative response to BMP6 in control PASMCs and the pro-proliferative response in HPAH PASMCs. Furthermore, *ACVR2A* siRNA (si*ACVR2A*) eliminated the enhanced TNFα/BMP6-dependent Smad1/5 phosphorylation (Fig. 3e), *ID1* induction (Fig. 3f) and proliferative responses (Fig. 3g) of HPAH PASMCs. TNFα enhanced *ACVR2A* expression in HPAH PASMCs without altering *ALK2*, *ALK3* or *ALK6* expression (Supplementary Fig. 10a,b). From these observations, we conclude that ALK2 mediates BMP6 signalling in PASMCs and TNFα-induced loss of BMPR-II permits preferential ACTR-IIA signalling, thus driving HPAH PASMC proliferation.

**TNFα alters NOTCH signalling in HPAH PASMCs**. The heightened Smad/ID signalling to BMP6/TNFα is unlikely to promote dPASMC proliferation since Smad/ID signalling inhibits PASMC proliferation[28,29] and reduced pulmonary vascular Smad/ID signalling is consistently reported in PAH in humans and animal models[5,39,40]. As NOTCH3 is implicated in PASMC hyperplasia in PAH[41,42] and TNFα induces the ADAM17-dependent cleavage of NOTCH[43,44], we examined the expression levels of the NOTCH family cleaved/transmembrane intracellular (NTM) regions and NOTCH family mRNA in control and HPAH dPASMCs. No significant difference was observed between control and HPAH PASMCs regarding the basal protein levels of the cleaved/transmembrane intracellular (NTM) regions

of NOTCH1, NOTCH2 and NOTCH3 (Supplementary Fig. 11a,b). Unexpectedly, TNFα increased the NOTCH1 and NOTCH2 NTM regions while reducing NOTCH3-NTM, the full-length NOTCH proteins being too faint to observe (Supplementary Fig. 11c–e). Although NOTCH expression does not necessarily represent signalling, the transcription of *NOTCH2* and its targets, *HEY1* and *HEY2* was increased, whereas *NOTCH3* and its target, *HES1*, were suppressed (Supplementary Fig. 12a–f and Supplementary Fig. 13a–g), implying NOTCH signalling was also altered. Since BMP6 enhanced these responses, we questioned whether BMPR-II and ACTR-IIA regulate NOTCH expression levels.

In control dPASMCs, si*BMPR2* enhanced TNFα-induced NOTCH2-NTM generation and *NOTCH2* transcription, with little effect on NOTCH1: BMP6 co-incubation accentuated the NOTCH2 response (Fig. 4a and Supplementary Fig. 14a,b). *BMPR2* silencing also promoted the TNFα-dependent reduction of NOTCH3-NTM generation and *NOTCH3* transcription in the presence of BMP6 (Fig. 4a and Supplementary Fig. 14c). In control dPASMCs, the TNFα-dependent *NOTCH2* induction following si*BMPR2* was inhibited by co-silencing with si*ACVR2A* (Supplementary Fig. 14b). Of note, si*ACVR2A* reduced BMP6-stimulated *NOTCH2* expression regardless of si*BMPR2* in control dPASMCs (Supplementary Fig. 14b). In *BMPR2* heterozygous HPAH PASMCs co-treated with TNFα and BMP6, si*ACVR2A* reduced NOTCH2-NTM generation and abrogated NOTCH3-NTM reduction (Fig. 4b), whereas TNFα alone had little effect. The effect on NOTCH1 protein was relatively weak in control and HPAH cells (Supplementary Fig. 14d,e). Collectively, these data indicate that TNFα, in particular when added with BMP6, regulates NOTCH2 and NOTCH3 expression in PASMCs via preferential ACTR-IIA signalling in BMPR-II-deficient cells.

Consistent with our *in vitro* data, *Notch2* mRNA expression and NTM levels were increased and *Notch3* decreased in the lungs, but not livers, of mice expressing *SP-C/Tnf* compared with control mice (Fig. 4c,d and Supplementary Fig. 14f–h). Moreover, concentric pulmonary arteriolar lesions in HPAH demonstrated similar NOTCH2 immunostaining compared with control vessels (Fig. 4e), but the increased number of PASMCs in the lesion is consistent with increased total NOTCH2 protein in the vasculature. In contrast, NOTCH3 levels were low in both (Fig. 4e).

**TNFα/BMP6 drive HPAH PASMC proliferation via NOTCH signals**. We addressed whether NOTCH signalling mediates the proliferative response of HPAH PASMCs to TNFα/BMP6. The γ-secretase inhibitor, DAPT, previously reported to inhibit PASMC proliferation through NOTCH3 blockade[41], both prevented the proliferative responses of HPAH PASMCs to TNFα and BMP6 (Supplementary Fig. 15a) and inhibited the anti-proliferative BMP6 response in control PASMCs (Supplementary Fig. 15b). *NOTCH2* siRNA reduced this proliferation of HPAH PASMCs to TNFα and BMP6, whereas *NOTCH3* siRNA did not (Fig. 4f,g). In control PASMCs, *NOTCH3* siRNA attenuated the anti-proliferative responses whereas *NOTCH2* siRNA had no effect (Fig. 4h,i). Consistent with the NOTCH2-dependent proliferation of HPAH dPASMCs to TNFα and BMP6, either *HEY1* or *HEY2* knockdown prevented this response (Supplementary Fig. 15c). Collectively, these data suggest that in HPAH is associated with loss of the anti-proliferative NOTCH3 pathway and gain of pro-proliferative NOTCH2 responses via HEY1 and HEY2.

**TNFα and BMP6 regulate NOTCH via c-SRC family kinases**. Since the kinase c-SRC integrates BMP and NOTCH signalling

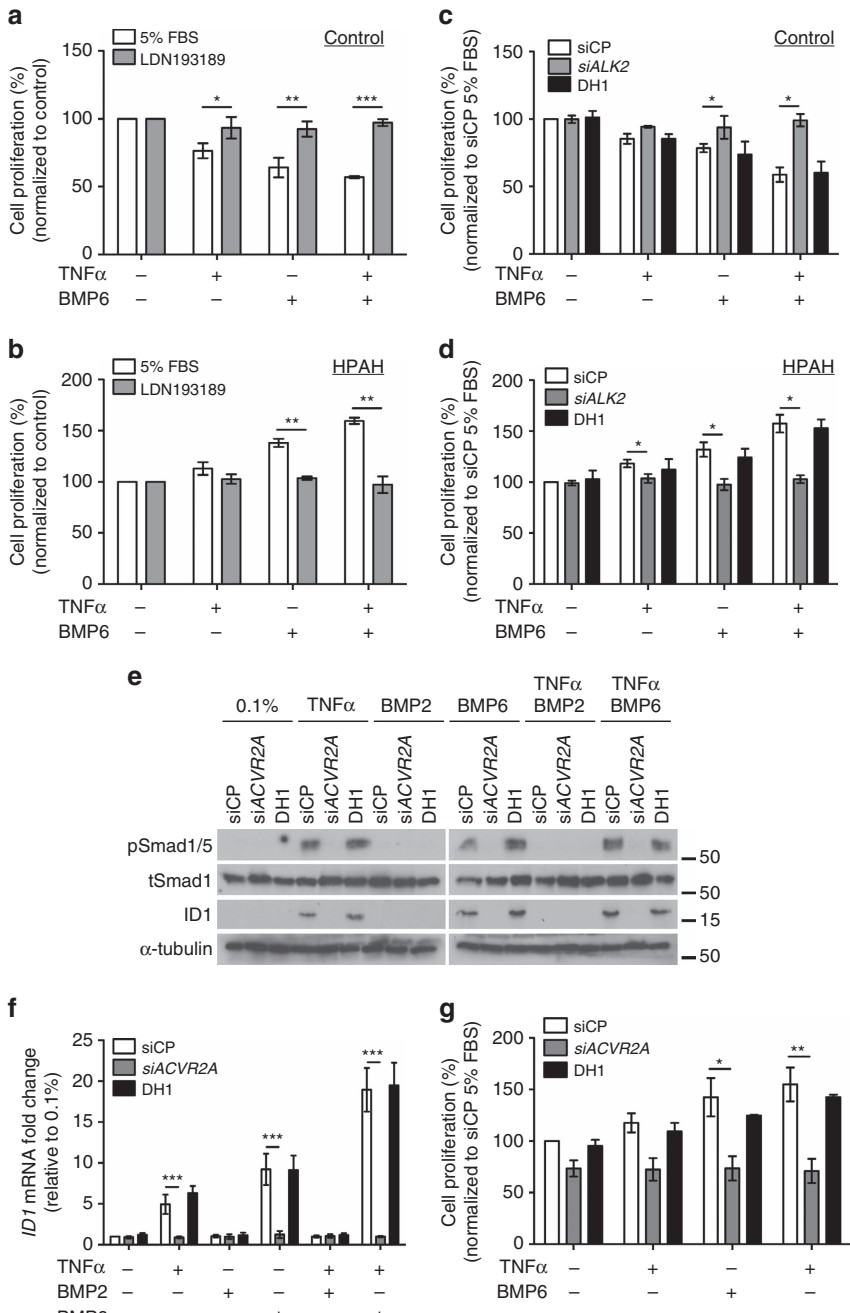

**Figure 3 | Altered BMP6 responses in HPAH PASMC are mediated via ALK2 and ACTR-IIA.** (**a,b**) Proliferation of human control (**a**) and HPAH (**b**) dPASMCs following 6 days treatment every 48 h with TNFα (1 ng ml$^{-1}$) and/or BMP6 (50 ng ml$^{-1}$) in the presence of LDN193189 (250 nM) ($n = 3$ control and HPAH cell lines). (**c,d**) Proliferation of human control (**c**) or HPAH (**d**) dPASMCs on day 6 following transfection with DharmaFECT1$^{TM}$ alone (DH1), si*ALK2* or non-targeting siRNA control (siCP) and treatment every 48 h with TNFα (1 ng ml$^{-1}$) and/or BMP6 (50 ng ml$^{-1}$) as indicated ($n = 3$ control and HPAH cell lines). (**e,f**) Human HPAH dPASMCs following transfection with DH1 alone, si*ACVR2A* or siCP and treated with TNFα (1 ng ml$^{-1}$) and/or BMP2 (10 ng ml$^{-1}$) or BMP6 (10 ng ml$^{-1}$) for 24 h as indicated. (**e**) Representative immunoblots of phospho-Smad1/5, total Smad1 and ID1 expression, reprobed for α-tubulin to ensure equal loading. The data shown are representative of three HPAH cell lines. (**f**) ID1 mRNA expression normalized to *ACTB* ($n = 3$). (**g**) Proliferation of human HPAH dPASMCs on day 6 following transfection with DH1 alone, si*ACVR2A* or siCP. Cells were treated every 48 h with TNFα (1 ng ml$^{-1}$) and/or BMP6 (50 ng ml$^{-1}$) as indicated ($n = 3$ HPAH cell lines). One-way analysis of variance with *post hoc* Tukey's for multiple comparisons used in **a,b,c,d,f** and **g**. *$P \leq 0.05$, **$P \leq 0.01$, ***$P \leq 0.001$. Error bars and mean ± s.e.m.

and has been implicated in PAH[27], we questioned whether the SRC family provided the mechanistic link between these pathways. SRC family activation was assessed through tyrosine-527 (Y527) dephosphorylation and tyrosine 416 (Y416) phosphorylation (Fig. 5a). HPAH PASMCs exhibited

SRC family activation to TNFα alone, or with BMP6, whereas control PASMCs did not (Fig. 5b). Importantly, si*BMPR2* transfection in control dPASMCs recapitulated the SRC activation to TNFα and BMP6 seen in HPAH PASMCs (Fig. 5c). Conversely, si*ACVR2A* abolished SRC activation

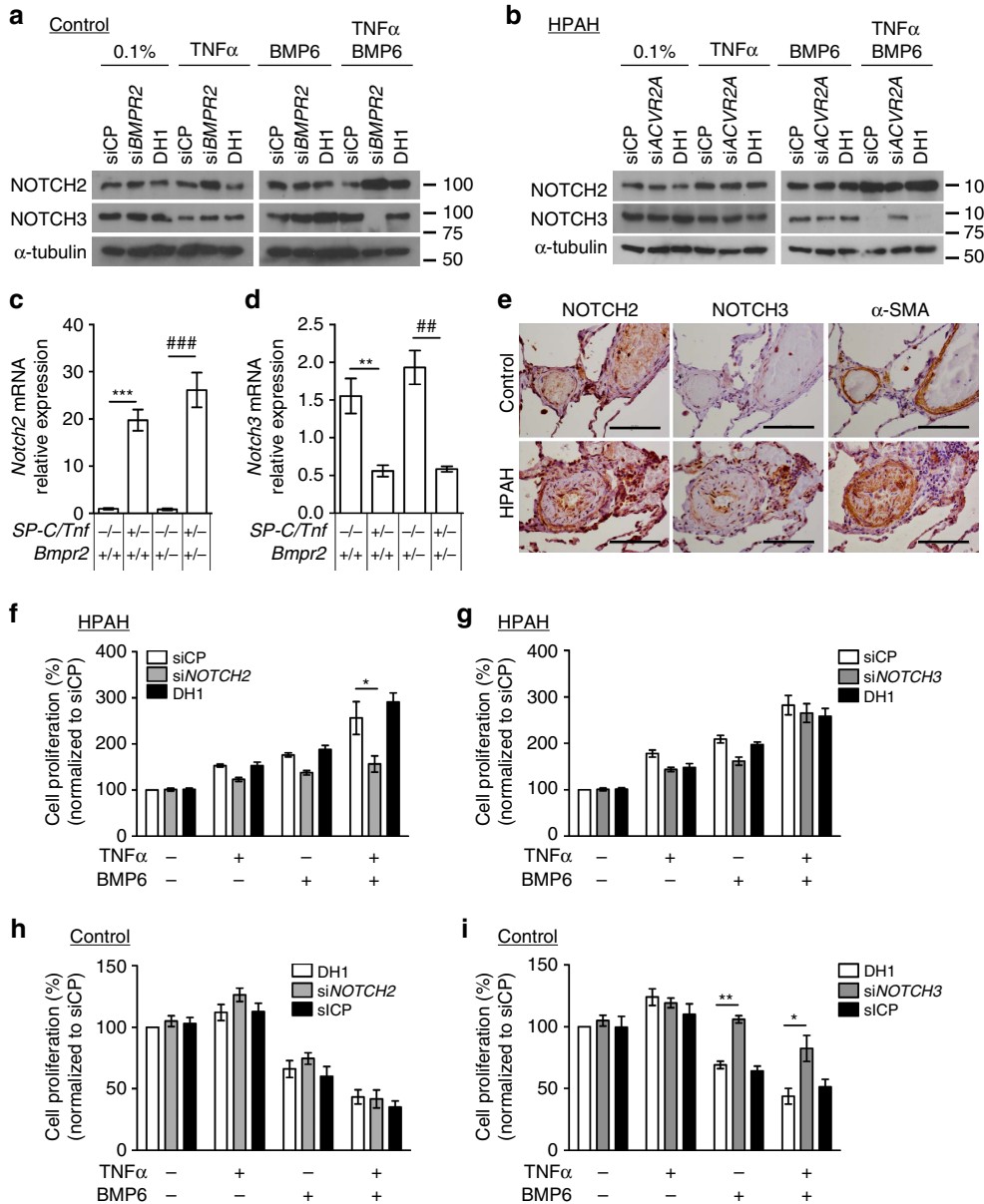

**Figure 4 | TNFα alters NOTCH expression.** (**a**) Representative immunoblots of cleaved/transmembrane intracellular (NTM) regions for NOTCH2 and NOTCH3 in human control dPASMCs following transfection with DharmaFECT1 alone (DH1), si*BMPR2*, or non-targeting siRNA control (siCP) and treatment with TNFα (1 ng ml$^{-1}$) and/or BMP6 (10 ng ml$^{-1}$) for 1 h as indicated. Reprobed for α-tubulin to ensure equal loading. The data shown are representative of three experiments. (**b**) Representative immunoblots of NTM regions for NOTCH2 and NOTCH3 in human HPAH dPASMCs following transfection with DH1 alone, si*ACVR2A*, or siCP and treated with TNFα (1 ng ml$^{-1}$) and/or BMP6 (10 ng ml$^{-1}$) for 1 h as indicated. Reprobed for α-tubulin to ensure equal loading. The data shown are representative of three experiments. (**c,d**) *Notch2* (**c**) and *Notch3* (**d**) mRNA expression in lungs isolated from 8-9 week old *Bmpr2*$^{+/+}$, *SP-C/Tnf/Bmpr2*$^{+/+}$, *Bmpr2*$^{+/-}$ and *SP-C/Tnf/Bmpr2*$^{+/-}$ ($n = 4$ per group) transgenic mice. Expression was normalized to *Actb*. (**e**) Representative images of immunohistochemical staining for NOTCH2, NOTCH3 and αSMA in lung sections from control and HPAH subjects. Scale bars, 100 μm. (**f,g**) Proliferation of human HPAH dPASMCs on day 6 following transfection with DH1 alone, si*NOTCH2* (**f**), si*NOTCH3* (**g**) or siCP and treatment every 48 h with TNFα (1 ng ml$^{-1}$) and/or BMP6 (50 ng ml$^{-1}$) as indicated ($n = 3$ cell lines). (**h,i**) Proliferation of human control dPASMCs on day 6 following transfection with DH1 alone, si*NOTCH2* (**h**), si*NOTCH3* (**i**) or siCP and treatment every 48 h with TNFα (1 ng ml$^{-1}$) and/or BMP6 (50 ng ml$^{-1}$) as indicated ($n = 3$ cell lines). One-way analysis of variance with *post hoc* Tukey's for multiple comparisons used in **c,d,f** and **i**. $*P \le 0.05$, $**/##P \le 0.01$, $***/###P \le 0.001$. Error bars represent mean ± s.e.m.

in HPAH PASMCs (Fig. 5d). Furthermore, the use of two pan SRC inhibitors in HPAH PASMCs abrogated the transcriptional induction of *NOTCH1* and *NOTCH2* and repression of *NOTCH3* by TNFα (Supplementary Fig. 16a–c). Collectively, these data suggest that TNFα, and to a lesser extent BMP6, activate SRC kinases to regulate *NOTCH1-3* in PASMCs.

SRC antibodies detect multiple family members, including FYN and YES, so we determined the contributions of individual members to the NOTCH responses in HPAH PASMCs using specific siRNAs (Supplementary Fig. 16d), particularly as SRC and FYN can interact with BMPR-II[27] and ACTR-IIA[45], respectively. In HPAH PASMCs, siRNA targeting *FYN* prevented the TNFα-dependent *NOTCH1* and *NOTCH2* induction

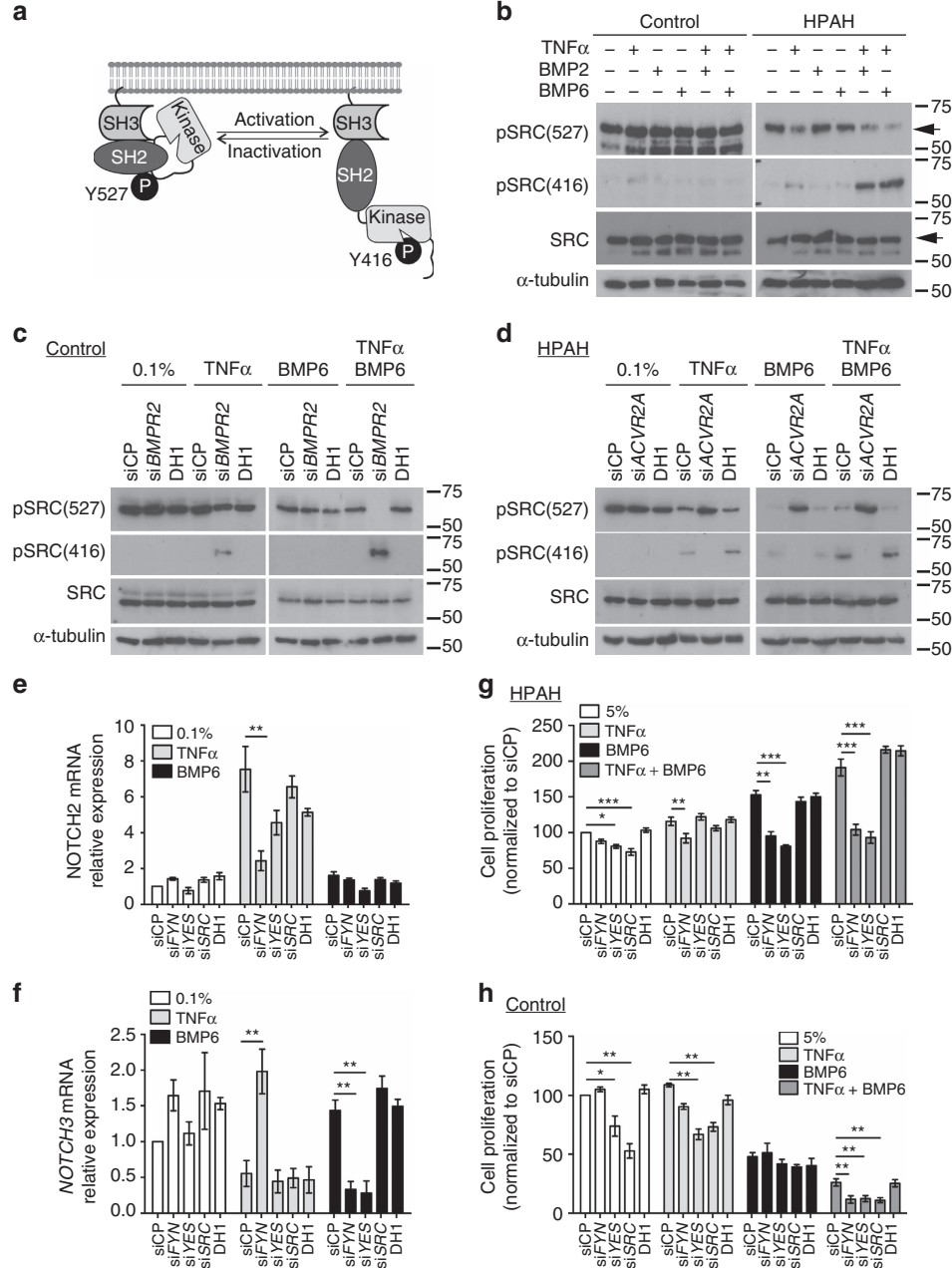

**Figure 5 | SRC kinases are activated by TNFα and BMP6, and can regulate Notch.** (**a**) Schematic depicting SRC phosphorylation. (**b**) Representative immunoblots of phospho-SRC Tyr527, Tyr416 and total SRC expression in human dPASMCs from disease-free controls and HPAH patients treated with TNFα (1 ng ml$^{-1}$) and/or BMP2 (10 ng ml$^{-1}$) or BMP6 (10 ng ml$^{-1}$) for 30 min as indicated. Reprobed for α-tubulin to ensure equal loading. The arrows indicate the SRC bands. The data shown are representative of three control and HPAH cell lines. (**c**) Representative immunoblots of phospho-SRC Tyr527, Tyr416 and total SRC expression in human control dPASMCs following transfection with DharmaFECT1 alone (DH1), si*BMPR2*, or non-targeting siRNA control (siCP) and treated with TNFα (1 ng ml$^{-1}$) and/or BMP6 (10 ng ml$^{-1}$) for 30 min as indicated. Reprobed for α-tubulin to ensure equal loading. The data shown are representative of three experiments. (**d**) Representative immunoblots of phospho-SRC Tyr527, Tyr416 and total SRC expression in human HPAH dPASMCs following transfection with DH1 alone, si*ACVR2A*, or siCP and treatment with TNFα (1 ng ml$^{-1}$) and/or BMP6 (10 ng ml$^{-1}$) for 30 min as indicated. Reprobed for α-tubulin to ensure equal loading. The data shown are representative of three cell lines. (**e,f**) *NOTCH2* and *NOTCH3* mRNA expression in human HPAH dPASMCs following transfection with DH1 alone, si*FYN*, si*YES*, si*SRC* or siCP and treatment with TNFα (1 ng ml$^{-1}$) or BMP6 (10 ng ml$^{-1}$) for 1 h as indicated. Expression was normalized to *ACTB* (n = 3). (**g**) Proliferation of human HPAH dPASMCs on day 6 following transfection with DH1 alone, si*FYN*, si*YES*, si*SRC* or siCP and treatment every 48 h with TNFα (1 ng ml$^{-1}$) and/or BMP6 (50 ng ml$^{-1}$) as indicated (n = 3). (**h**) Proliferation of human control dPASMCs on day 6 following transfection with DH1 alone, si*FYN*, si*YES*, si*SRC* or siCP and treatment every 48 h with TNFα (1 ng ml$^{-1}$) and/or BMP6 (50 ng ml$^{-1}$) as indicated (n = 3). One-way analysis of variance with *post hoc* Tukey's for multiple comparisons used in **e,f,g** and **h**. *$P \leq 0.05$, **$P \leq 0.01$, ***$P \leq 0.001$. Error bars represent mean ± s.e.m.

and *NOTCH3* repression (Fig. 5e,f and Supplementary Fig. 16e) whereas either FYN or YES mediated the repression of *NOTCH3*

by BMP6 (Fig. 5f). *FYN* or *YES* siRNAs also abolished the proliferative response to TNFα or BMP6 alone, or in combination

(Fig. 5g). In control PASMCs, loss of YES or SRC reduced serum-dependent proliferation but minimally impacted on the BMP6 and TNFα responses (Fig. 5h and Supplementary Fig. 16f).

**Anti-TNFα reduces PAH and aberrant TNF/BMP signals *in vivo*.** Having demonstrated that TNFα subverts BMP signalling and drives PASMC proliferation via c-SRC and NOTCH2, we examined this in the rat Sugen-hypoxia (S/H) model of PAH[46] (Supplementary Fig. 17a). We also explored the impact of the anti-TNFα therapeutic, etanercept (soluble TNFR-II conjugated to human IgG-Fc) on established PAH. Exposure of rats to S/H induced robust PAH (Fig. 6a,b)

associated with pulmonary vascular remodelling (Fig. 6c and Supplementary Fig. 17b). Etanercept reversed the progression of PAH, reducing RVSP, right ventricular hypertrophy and muscularization of small alveolar duct-associated arterioles (Fig. 6a-c), but not the wall thickness of the larger arterioles associated with terminal bronchioles (Supplementary Fig. 17b), without altering left ventricular function (Supplementary Table 1). The development of PAH in the S/H model was associated with BMP and NOTCH signalling changes consistent with our *in vitro* data. *Bmpr2* expression and Smad1/5 signalling were reduced (Fig. 6d,e) and *Acvr2a*, *Alk2*, *Bmp6* and *Tnf* expression were all increased (Fig. 6d and Supplementary Fig. 17c,d). Also, *Notch1*, *Notch2*, *Hey1* and *Hey2*

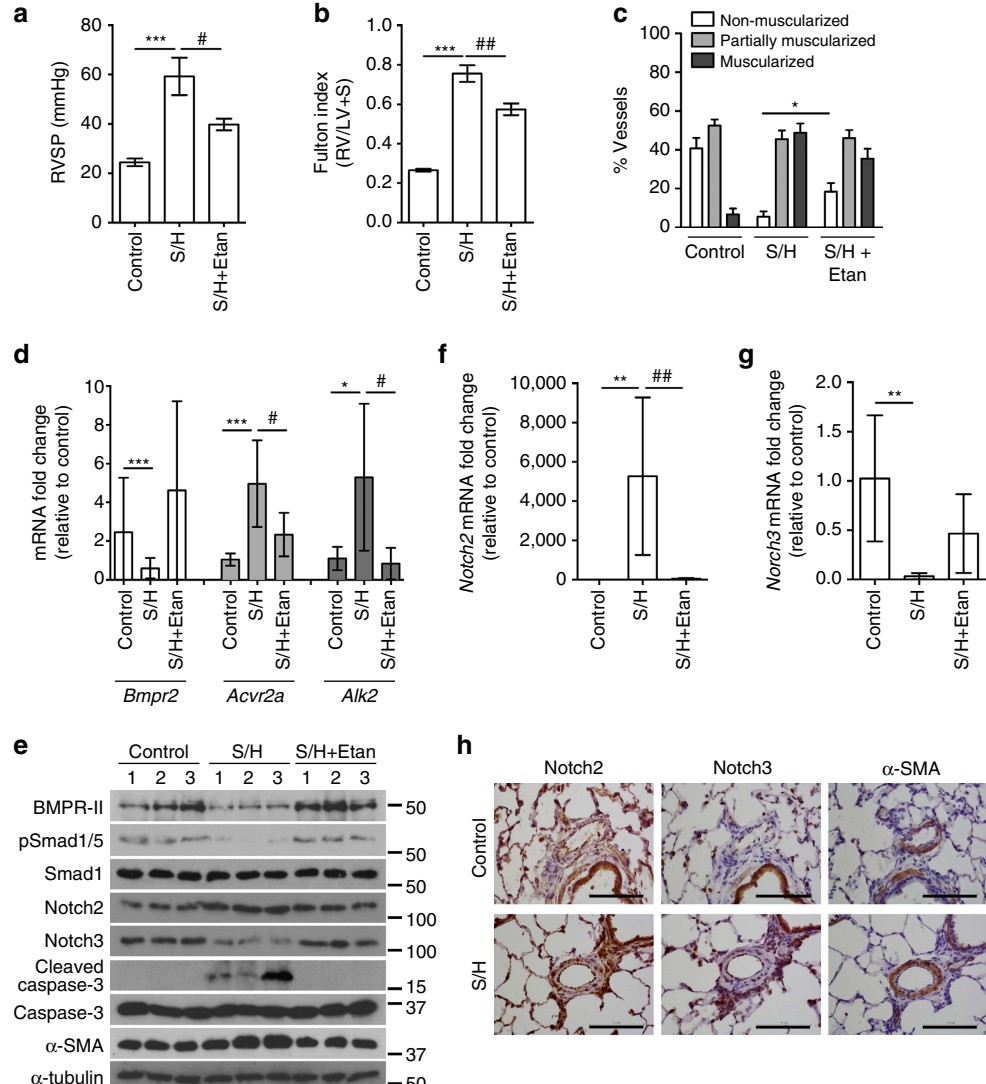

**Figure 6 | Etanercept reverses established pulmonary hypertension in the Sugen-hypoxia rat model.** Rats were given vehicle injections and maintained in normoxia (Control, $n = 6$) or challenged with SU-5416 (20 mg kg$^{-1}$, s.c.) and 3 weeks of hypoxia (10% O2) before 5 weeks of normoxia and 3 weeks of biweekly treatment with saline vehicle (S/H, $n = 9$) or etanercept (S/H + Etan, $n = 9$; 2.5 mg kg$^{-1}$, i.p.). (**a,b**) Assessment of RVSP (**a**) and right ventricular hypertrophy (Fulton index (RV/LV + S)) (**b**). (**c**) Quantification of non-, partially and fully muscularized arteries as a percentage of total alveolar wall and duct arteries ($n = 6$ for control, $n = 9$ for all other groups; Student's *t*-test for non-muscularized vessels). (**d**) BMPR2, ACVR2A and ALK2 mRNA expression, normalized to *Actb*, in lungs isolated from control, S/H and S/H + Etan rats ($n = 6$). (**e**) Representative immunoblots of BMPR-II, phospho-Smad1/5, total Smad1, Notch2, Notch3, cleaved Caspase3, total Caspase3 and αSMA expression in lungs isolated from control, S/H and S/H + Etan rats. Reprobed for β-actin to ensure equal loading ($n = 3$). (**f,g**) *Notch2* and *Notch3* mRNA expression in lungs isolated from lungs of control, S/H and S/H + Etan rats. Expression was normalized to *Actb* ($n = 6$). (**h**) Representative images of immunohistochemical staining for Notch2, Notch3 and αSMA in lung sections from control and S/H rats. Scale bars, 100 μm. One-way analysis of variance with *post hoc* Tukey's for multiple comparisons used in **a,b,d,f** and **g**. */#$P \leq 0.05$, **/##$P \leq 0.01$, ***$P \leq 0.001$. Error bars represent mean ± s.e.m.

expression and medial *Notch2* staining were increased in S/H rats (Fig. 6e,f,h and Supplementary Fig. 17e,f) whereas *Notch3* and *Hes1* were reduced (Fig. 6e,g and Supplementary Fig. 17g). Furthermore, we observed increased caspase-3 cleavage (endothelial apoptosis) and alpha smooth muscle actin expression (muscularization; Fig. 6e). In S/H animals treated with etanercept, the reversal of PAH progression was associated with restored BMPR-II, phospho-Smad1/5 and *Notch3* expression (Fig. 6d,e,g) and a reduction of the pathological increases in *Acvr2a*, *Alk2*, *Bmp6*, *Tnf*, *Notch2*, *Hey1*, *Hey2*, cleaved caspase-3 and alpha-smooth muscle actin (Fig. 6d-f and Supplementary Fig. 17 c,d,f). These observations support our contention that increased TNFα signalling in PAH causes an imbalance of BMP and NOTCH signalling that can be redressed through therapeutic targeting of the TNFα pathway.

## Discussion

The mechanisms by which loss-of-function *BMPR2* mutations underlie severe PAH with low penetrance have remained elusive. Here we provide novel mechanistic insights into a critical interaction whereby TNFα drives the development of PAH by repressing vascular *BMPR2* transcription and promoting BMPR-II cleavage in PASMCs. The redundancy between ADAM10 and ADAM17 for BMPR-II cleavage is intriguing. We suggest that this might permit BMPR-II cleavage only in cells expressing both ADAMs, restricting this response compared with the shedding of other molecules by the individual ADAMs. Ultimately, the impact of severe BMPR-II reduction combined with enhanced BMP6 signalling via ALK2/ACTR-IIA and c-SRC family members, promotes PASMC proliferation through aberrant NOTCH2/3 expression and their downstream transcriptional targets. Furthermore we confirm these alterations in the hypertensive lungs of PAH patients and preclinical rodent PAH models. The observation that etanercept treatment in preclinical PAH normalized BMPR-II levels, restored normal NOTCH expression levels and reversed the progression of PAH provides a justification to explore the clinical use of anti-TNFα approaches in PAH patients.

Inflammatory cytokines are associated with the pathogenesis of PAH[8,9,12] and we demonstrate that local TNFα expression is present in the medial layers of pulmonary arteries from IPAH and HPAH patients, but not in control tissues. Given that we previously reported elevated systemic circulating TNFα levels of 10.45 pg ml$^{-1}$ in IPAH patients and 9.85 pg ml$^{-1}$ in HPAH patients compared with 7.92 pg ml$^{-1}$ in controls[12], we suggest that local lung expression is likely to generate appreciably higher TNFα levels and have a more restricted effect on the pulmonary circulation than small elevation in these relatively low systemic levels. Our demonstration that TNFα suppressed BMPR-II in pulmonary vascular cells confirms reports in osteoblasts[47] and aortic endothelial cells[19]. Furthermore, TNFα exacerbates the genetic *BMPR2* haploinsufficiency in HPAH PASMCs causing the substantial reduction of BMPR-II levels that allow BMP6 to switch signalling to the alternative type II receptor, ACTR-IIA[21]. Also, TNFα increased BMP6 expression in PASMCs and hPAECs and induced *ACVR2A* expression in HPAH PASMCs. The resulting BMP6/ALK2/ACTR-IIA utilization induced paradoxical increases in transient Smad1/5 responses, characteristic of BMP receptor complex switching[30,31]. Since PAH is associated with reduced Smad signalling[39,40], this transient Smad response was unlikely to promote heightened PASMC proliferation, so other candidate pathways were considered.

Emerging evidence implicates NOTCH in the pathogenesis of PAH. NOTCH inhibition by soluble JAGGED1 attenuates PAH in hypoxic and MCT-PAH rat models[48]. Our data suggest that, on the background of BMPR-II haploinsufficiency, inappropriate NOTCH2 expression and downstream HEY1/2 signalling in response to TNFα stimulate PASMC proliferation. NOTCH2 is abundantly expressed in vascular SMCs and NOTCH2 deletion reduces SMC number and causes embryonic lethality[49,50]. The reduction of NOTCH3 expression and HES1 signalling we observed was surprising given previous reports of NOTCH3 promoting PAH[41,51,52]. However, these previous studies focussed on NOTCH3 and used DAPT as the therapeutic intervention in PAH models, which blocks NOTCH2 and NOTCH3 cleavage. DAPT inhibited the proliferative responses of our HPAH PASMCs, so NOTCH2 blockade will also be affected in these previous reports. A previous study demonstrated that NOTCH3 knockout mice are protected from hypoxia-induced pulmonary hypertension[41]. We would contest whether this observation supports a specific role for NOTCH3 in the development of pulmonary hypertension, since *Notch3* knockout mice demonstrate grossly abnormal arterial maturation in all vascular beds with altered myogenic responses and structural defects[53]. In view of this, it is not surprising that a mouse deficient in *Notch3* during development exhibits a deficient response to chronic hypoxic exposure. We observed low baseline NOTCH3 staining in the pulmonary arterial media, so further reduction in PAH was not obvious. Although the intensity of NOTCH2 staining was similar in control and HPAH vessels, proliferation of cells in the vascular media represented an increase in the number of NOTCH2-positive cells. Our data imply that NOTCH2, via HEY1/HEY2, enhances HPAH cell proliferation, while NOTCH3 appears critical in suppressing PASMC proliferation. We acknowledge that our data contradicts some aspects of previous reports regarding the contribution of NOTCH signalling in PAH, but our studies primarily focus on the combined impact of TNFα and a background of *BMPR2* haploinsufficiency, which is the context relevant to the majority of human heritable cases.

We explored the role of SRC family kinases in linking the TNFα, BMP and NOTCH pathways. As BMPR-II sequesters c-SRC and renders it inactive following BMP stimulation[27], we hypothesized that BMPR-II reduction increases the availability of SRC kinases to interact with ACTR-IIA or TNFα receptors. In this context, we identified FYN as a key regulator of the aberrant NOTCH2 expression and proliferation to TNFα and a dual role for FYN and YES in the proliferative response to TNFα and BMP6. To date, only one report has identified an interaction of FYN and ACTR-IIA in neuronal cells[45], so our study is the first to identify the roles of specific SRC members in the HPAH PASMC proliferative response, reminiscent of the constitutive activation of these proto-oncogenes in carcinogenesis[54].

In summary, TNFα induces BMP6 and exacerbates the reduced BMPR-II expression in HPAH PASMCs, enabling BMP6 to recruit the ALK2/ACTR-IIA receptor complex. TNFα promotes excessive PASMC proliferation via activation of FYN and the NOTCH2-HEY1/2 axis, while simultaneously suppressing the anti-proliferative NOTCH3-HES1 axis. Collectively, these findings provide a mechanism by which inflammatory TNFα signalling promotes pulmonary vascular remodelling in the setting of BMPR-II deficiency in PASMCs. We acknowledge that TNFα may also impact on endothelial cells to promote the pathogenesis of PAH. Indeed, we show that TNFα represses BMPR-II in hPAECs and a previous study reported that TNFα promotes granulocyte macrophage-colony stimulating factor (GM-CSF) secretion and macrophage recruitment in BMPR-II-deficient endothelial cells[55]. Also, administration of anti-DLL4, which primarily targets endothelial cell NOTCH1 signalling,

causes PAH in patients with advanced solid tumours[56]. Therefore, the impact of TNFα on endothelial NOTCH expression and signalling warrants additional investigation. Such mechanisms may be responsible for disease penetrance in patients carrying mutations in BMPR-II. Moreover, our findings justify the testing of anti-TNFα approaches in treatment of PAH.

## Methods

**Cell culture and treatments.** Human pulmonary artery endothelial cells (PAECs) were purchased from Lonza (Cat. No. CC-2530; Basel, Switzerland) and maintained in EGM-2 with 2% FBS (Lonza), as per the supplier's instructions and were used between passages 4 and 8. For experimental studies, cells were starved overnight in Medium 199 containing 0.1% FBS and Antibiotic-Antimycotic (A/A; 100 U ml$^{-1}$ penicillin, 100 mg ml$^{-1}$ streptomycin and 0.25 mg ml$^{-1}$ amphotericin B, Invitrogen, Renfrewshire, UK) and incubated in 2% foetal bovine serum (FBS) containing A/A without growth factors overnight. Cell lines were routinely tested for mycoplasma contamination and only used if negative.

Distal human pulmonary artery smooth muscle cells (dPASMCs) were derived from small vessels ($<2$ mm diameter) lung resection specimens. The details of the patients from whom the cells were derived are provided in Supplementary Table 2. The lung parenchyma was dissected away from a pulmonary arteriole, following the arteriolar tree, to isolate 0.5-to 2-mm-diameter vessels. These were dissected out and cut into small fragments, which were plated in T25 flasks and left to adhere for 2 h. A section of the pulmonary arteriole was collected, fixed in formalin and embedded in paraffin, and sections were analysed to ensure that the vessel was of pulmonary origin.

Proximal PASMCs were isolated from vessel segments (5-8 mm diameter) that were cut to expose the luminal surface. The endothelium was removed by gentle scraping with a scalpel blade and the media then peeled away from the underlying adventitial layer. The medial explants were cut into 4- to 9-mm$^2$ sections, plated into T25 flasks and allowed to adhere for 2 h before addition of DMEM (Invitrogen) containing 20%(v/v) FBS and A/A (DMEM/20%FBS). Papworth Hospital ethical review committee approved the use of the human tissues (Ethics Ref 08-H0304-56 + 5) and informed consent was obtained from all subjects.

For isolation of peripheral lung rat PASMCs, rats were anaesthetized with pentobarbitone sodium (Sagatal, 60 mg kg$^{-1}$ i.p.), a small incision made in the neck and the trachea cannulated. The chest was opened and the pulmonary artery cannulated with a metal cannula introduced through the right ventricle. An incision was made in the left atrium and the lungs rinsed via the pulmonary artery catheter using warm PBS (10 ml) with Heparin (500U ml$^{-1}$). Prewarmed (37 °C) 0.5% Ferric oxide (Fe3O4, cat.no. 310069, Sigma-Aldrich, Gillingham, Dorset, UK)/1% agarose in DMEM was slowly injected via the PA cannula followed by injection of 1% agarose in DMEM via the tracheal cannula. The lung block was excised and transferred to DMEM on ice for about 20 min to set the agarose. The lungs were sliced with subsequent trimming of the subpleural margin to a depth of no more than 1mm and the slices minced with a razor blade and partially digested with type II collagenase (80 U ml$^{-1}$ of culture medium) for 30 min at 37 °C. The tissue was sheared through 18 gauge needles, transferred to $12 \times 75$-mm glass tubes, placed in a magnetic separator (Promega, Madison, WI) and rinsed with DMEM (4 °C) until a clear discard solution was obtained. The microvessels were isolated magnetically, resuspended in 1 ml of DMEM/20%FBS, plated in a 25 cm$^2$ tissue culture flask, and incubated in humidified air with 5% CO$_2$ at 37 °C. On day 2, an additional 4 ml of DMEM/20%FBS was added to each flask. Once confluent, the cells were detached with trypsin (0.05%) and 0.02% EDTA in Hanks' balanced salts (HBSS) and cultured in DMEM/20%FBS. Mouse PASMCs were isolated from explants of small pulmonary arteries (0.5 mm internal diameter) microdissected from wild-type mice.

Once adhered, tissue explants were incubated in DMEM/20% FBS until cells had grown out and were forming confluent monolayers. PASMCs were trypsinized, and subsequent passages were propagated in DMEM supplemented with 10% heat-inactivated FBS and A/A (DMEM/10% FBS) and maintained at 37 °C in 95% air-5% CO$_2$. The smooth muscle phenotype was confirmed by positive immunofluorescent staining using an antibody to smooth muscle specific alpha-actin (Clone IA4 Sigma-Aldrich; 1:100 dilution). Human aortic smooth muscle cells isolated from patients providing informed written consent, under local ethics approval (NRES Committee East of England – Norfolk). Aortic cells were kindly provided by Dr Murray Clarke (University of Cambridge, UK). All smooth muscle cell lines were used between passages 4 and 8.

C2C12-BRE cells, a subcloned population of C2C12 myoblasts stably expressing Smad-responsive BMP response element (BRE) generated in the laboratory of Dr G.Inman, were cultured in DMEM (10% FBS, 2 mM L-glutamine and 700 µg ml$^{-1}$ G418)[57].

Smooth muscle cells and C2C12-BRE cells were quiesced in DMEM containing 0.1% FBS and A/A overnight before treatments. Recombinant human TNFα, BMP2, BMP4, BMP6, IL-1β, IL-6 and IL-8 were purchased from R&D Systems (Oxfordshire, UK). Recombinant mouse TNFα was purchased from PeproTech

(Rocky Hill, NJ). Cells were treated with TNFα (1 ng ml$^{-1}$) for 24 h, unless otherwise indicated. Cells were stimulated with BMP2, BMP4 or BMP6 (10 ng ml$^{-1}$ unless otherwise stated) or co-stimulated with TNFα and BMP ligand as indicated. The metalloprotease inhibitor batimastat (BB94) (10 ng ml$^{-1}$) was a kind gift from Dr. Murray Clarke (University of Cambridge, UK). ADAM10 inhibitor GI254023X (10 µM) was a kind gift from Prof. Andreas Ludwig (RWTH AACHEN, Germany)[58]. The anti-ADAM17 antibody, D1(A12) (50 nM) was a kind gift from Prof Gillian Murphy (Cancer Research UK Research Institute, Cambridge, UK)[59]. ADAM10/17 inhibitor TAPI-1 (10 µM) was purchased from Enzo Life Sciences (Devon, UK). The BMP signalling inhibitor, LDN193189, was a kind gift from Dr. Paul Yu (Brigham and Women's Hospital, Boston, MA). The γ-secretase inhibitor, DAPT, was from Sigma-Aldrich. Unless otherwise stated, cells were pretreated with inhibitors for 30 min before TNFα stimulation and then added to cells for a total of 24 h. For immunoneutralization studies, treatments were preincubated with 1 µg ml$^{-1}$ monoclonal anti-BMP2 (MAB3551, R&D Systems) or anti-BMP6 (MAB507, R&D Systems) for 1 h before adding to cells.

**Cell proliferation.** For assessment of PASMC proliferation, cells were seeded at 15,000 cells per well in 24-well plates and left to adhere overnight. After 48 h, cells were washed with DMEM containing 0.1% FBS and A/A and then serum-restricted in DMEM containing 0.1% FBS and A/A for 16 h. Cells were then exposed to the stated treatments in DMEM containing 5% FBS and A/A and treatments were replenished every 48 h. At the relevant time points, cells were trypsinized and counted on a hemocytometer using trypan blue exclusion to assess cell viability.

**Expression plasmids and reagents.** The pcDNA3 expression plasmid encoding 5′-myc-tagged BMPR-II wild type was prepared as previously described[60]. Mutant myc-tagged BMPR-II V158A, V160A, V163A and V166A plasmids were created using the QuikChange Site-Directed Mutagenesis kit (Agilent Technologies, Cheshire, UK) according to the manufacturer's instructions (Supplementary Table 3). The presence of each mutation was verified by sequencing.

**Adenoviral transduction.** The replication incompetent serotype 5 adenoviral plasma vectors, AdCMVBMPR2myc[61] and kinase-dead AdCMVBMPR2(D485G)myc[61], were digested with Pac1 and transfected into HEK293T cells. Large-scale virus preps were generated in HEK293 cells and purified by cesium chloride centrifugation. Viral titer was determined by TCID50 assay and particle titer by OD260. Cells were infected with 50 plaque-forming units (pfu) per cell for 4 h in serum-free DMEM and this was then replaced with DMEM/10% FBS for 16 h. Before treatment cells were serum-restricted in DMEM containing 0.1% FBS and A/A for 16 h and then treated with ligands in DMEM containing 5% FBS and A/A, with replenishment of ligands every 48 h. Cells were trypsinized and counted on day 6.

**Plasmid transfections.** Plasmids were prepared using the PureLink maxiprep kit (Invitrogen), according to the manufacturer's instructions. Before transfection, PASMCs were incubated with Opti-MEM-I (Invitrogen) for 2 h. Cells were transiently transfected with 4 µg of expression plasmid using 2 µl Lipofectamine 2000 reagent (Invitrogen) in Opti-MEM-I. Cells were incubated with transfection mixes for 4 h, followed by replacement with DMEM/10% FBS for 48 h before quiescence and treatment as indicated. Transfection efficiency was confirmed via BMPR-II and Myc tag immunoblotting.

**siRNA transfections.** Before transfection, PASMCs were incubated with Opti-MEM-I serum-free medium (Invitrogen) for 3 h before adding 10 nM siRNA lipoplexed with DharmaFECT1 (GE Dharmacon, Lafayette, CO) siRNA/Dhar-maFECT1 complexes were allowed to form for 20 min at room temperature before being added to the cells. Cells were then incubated with the complexes for 4 h at 37 °C before returning to DMEM/10% FBS overnight. Knockdown efficiency was confirmed by immunoblotting or mRNA expression. The siRNAs used were: ON-TARGETPlus Smartpool oligos for ($>x$% values represent knockdown at RNA level): ACVR2A ($>73$%), ADAM10, ADAM17, ALK2 ($>84$%), BMPR2 ($>75$%), RELA ($>82$%, encoding NF-κB p65) or a non-targeting control pool (siCP) (all GE Dharmacon) or oligos targeting FYN ($>64$%, SASI_Hs01_00195124), HEY1 ($>63$%, SASI_Hs01_0052320), HEY2 ($>63$%, SASI_Hs02_00343977), NOTCH2 ($>50$%, SASI_Hs01_00068801), NOTCH3 ($>75$%, SASI_Hs01_00101287), SRC ($>62$%, SASI_Hs01_00112907) or YES ($>68$%, SASI_Hs01_00086922) from Sigma-Aldrich. For proliferation experiments, we confirmed that the level of knockdown was similar at days 2, 4 and 6 for each target.

**Immunoblotting.** Frozen liver and lung tissue were homogenized in lysis buffer (250 mM Tris-HCl, pH 6.8, 4% SDS, 20% v/v glycerol, EDTA-free protease inhibitor cocktail (Roche, West Sussex, UK)) sonicated and centrifuged for 15 min at 15,000$g$. PAECs and PASMCs were snap-frozen on an ethanol-dry ice bath in lysis buffer (125 mM Tris (pH 7.4), 2% SDS, 10% glycerol, and EDTA-free

protease inhibitor cocktail). Cell lysates (20–100 µg protein) were separated by SDS-PAGE and proteins transferred to polyvinylidene fluoride membranes by semidry blotting (GE Healthcare, Buckinghamshire, UK). Membranes were then blocked and probed with rabbit polyclonal antibodies toward total Smad1 (cat. #9743; 1:1,000 dilution), phosphorylated SRC(Y527) (cat. #2105; 1:1,000 dilution) (all Cell Signaling Technology, Danvers, MA), ADAM10 (cat.no. ab1997; 1:1,000 dilution), ADAM17 (cat.no. ab39162; 1:1,000 dilution) Abcam, Cambridgeshire, UK); rabbit monoclonal antibodies toward phosphorylated Smad1/5 (cat. #9516; clone 41D10; 1:1,000 dilution), caspase-3 (cat. #9665; clone 8G10; 1:1,000 dilution), cleaved caspase-3 (cat. #9664; clone 5A1E; 1:1,000 dilution), NOTCH1 (cat. #3608; clone D1E11; 1:1,000 dilution), NOTCH2 (cat. #5732; clone D76A6; 1:1,000 dilution), NOTCH3 (cat. #5276; clone D11B8; 1:1,000 dilution), phospho-SRC(Y416) (cat. #6943; clone D49G4; 1:1,000 dilution), SRC (cat. #2123; clone 32G6, all Cell Signaling Technology, Danvers, MA), ID1 (cat.no. M085; clone 195-14, CalBioreagents, San Mateo, CA; 1:1,000 dilution); or mouse monoclonal antibodies towards BMPR-II (cat.no. BD612292; clone 18/BMPR-II, BD Transduction Laboratories, Franklin Lakes, NJ; 1:250 for SMCs and 1:400 for PAECs), c-Myc (cat.no. sc40; clone 9E10, Santa Cruz Biotechnology, Dallas, TX; 1:1,000 dilution). After washing, blots were incubated with secondary anti-mouse horseradish peroxidase (HRP) antibody (cat.no. P0447; Dako, Cambridgeshire, UK; 1:2,000 dilution) or anti-rabbit horseradish peroxidase antibody (cat.no. P0448; Dako; 1:2,000 dilution) for 1 h at room temperature. As a loading control, all blots were re-probed with a mouse monoclonal antibody toward either α-tubulin (T6199; clone DM1A, Sigma-Aldrich; 1:5,000 dilution and 1:5,000 anti-mouse HRP) or β-actin (A5441; clone AC-15, Sigma-Aldrich; 1:7,500 dilution and 1:7,500 anti-mouse HRP). Densitometry was performed using ImageJ software. Membranes were developed using enhanced chemiluminescence (GE Healthcare). Uncropped western blots are presented in Supplementary Figs 18–39.

**Deglycosylation.** Protein was deglycosylated using PNGase F according to the manufacturer's instructions (New England Biolabs, Hitchin, Hertfordshire, UK). Approximately 60–80 µg of protein was deglycosylated and then fractionated by SDS–PAGE.

**Immunoprecipitation.** Conditioned media were taken from transfected PASMCs before lysis. Media were centrifuged at 2,000g (4 °C) to remove cellular debris and stored in 1 ml aliquots at −80 °C. For immunoprecipitation, 1 ml culture media were incubated with a mouse monoclonal toward BMPR-II (cat.no. MAB811; Clone # 73805, R&D Systems; final dilution = 8 µg ml$^{-1}$) overnight on a rotary mixer at 4 °C. Antibody:protein complexes were isolated by incubation with protein-G sepharose beads (Sigma-Aldrich) for 4 h on a rotary mixer at 4 °C. Beads were washed three times in PBS/Triton X-100 (0.1%; Sigma-Aldrich) and resuspended in 2× loading buffer containing 62.5 mm Tris–HCL, pH 6.8, 10% (v/v) glycerol, 2% (w/v) SDS, 5% (v/v) β-mercaptoethanol, 0.003% (w/v) bromophenol blue. To detach complexes from the beads, samples were boiled at 99 °C for 5 min and centrifuged at 2,500g for 2 min at 4 °C. Samples were separated by SDS-PAGE and immunoblotted for myc tag as previously described.

**sBMPR-II ELISA.** Conditioned media, taken from PASMCs stimulated with TNFα, were centrifuged at 2,000 × g (4 °C) to remove cellular debris and stored in 1 ml aliquots at −80 °C. BMPR-II ECD was measured using an in-house ELISA. ELISA was performed as previously described with the following modifications[62]. Briefly, flat-bottom high binding 96-well ELISA plates (Greiner, South Lanarkshire, UK) were coated with 1 µg ml$^{-1}$ mouse monoclonal anti-human BMPR-II antibody (R&D Systems) for 2 h at room temperature. After washing, plates were blocked with 1% BSA (Sigma-Aldrich) in PBS-T for 1 h at room temperature. Aliquots of standards (His-tagged BMPR-II-ECD, Sino Biologicals, Beijing, China) and conditioned media with the relevant controls were added and incubated in a humidified chamber overnight at 4 °C. After washes, polyclonal rabbit anti-human BMPR-II antibody (Santa Cruz Biotechnology), diluted to 0.5 µg ml$^{-1}$ in 1% BSA/PBS-T, was added and incubated for 3 h at room temperature. Plates were washed as described and goat anti-rabbit alkaline phosphatase conjugate whole molecule IgG (Sigma-Aldrich) was added at a 1:1,000 dilution in 1% BSA/PBS-T, and incubated for a further 2 h at room temperature. Plates were developed and read at 405 nm in an automated plate reader (3550, Bio-Rad). Results were analysed using Microplate Manager software (Bio-Rad, Hemel Hempstead, Hertfordshire, UK). The lower level sensitivity of this assay was 750 pg ml$^{-1}$.

**sBMPR-II ligand-binding assay.** C2C12-BRE cells were treated with BMP2 or BMP4 (at 1 and 10 ng ml$^{-1}$) in the presence or absence of commercially available glycosylated His-tagged BMPR-II-ECD (Sino Biologicals) or conditioned media from transfected PASMCs stimulated with TNFα. Luciferase activity in the cells was assessed using a luciferase reporter assay kit (Roche).

**RNA preparation and quantitative reverse transcription–PCR.** Total RNA was extracted using the RNeasy Mini Kit with DNAse digestion (Qiagen, West

Sussex, UK). cDNA was prepared from ~1 µg of RNA using the High Capacity Reverse Transcriptase kit (Applied Biosystems, Foster City, CA), according to the manufacturer's instructions. All quantitative PCR reactions were prepared in MicroAmp optical 96-well reaction plates (Applied Biosystems) using 50 ng µl$^{-1}$ cDNA with SYBR Green Jumpstart Taq Readymix (Sigma-Aldrich), ROX reference dye (Invitrogen) and custom sense and anti-sense primers (all 200 nM). Primers for human: *ACTB* (encoding β-actin), *ADAM10*, *ADAM12*, *ADAM15*, *ADAM17*, *ALK3*, *BMPR2*, *HES1*, *HEY1*, *HEY2*, *ID1*, *MMP14*, *MMP15*, *MMP16*, *MMP17*, *MMP24*, *NOTCH3*; mouse *Acvr2a* (encoding Actr-IIa), *Alk2*, *Bmpr2*, *Notch1*, *Notch2* and *Notch3* were all designed using Primer3 (http://primer3.sourceforge.net/) (Supplementary Tables 4–6). QuantiTect primer assays (Qiagen) for: human *ACVR2A* (encoding ACTR-IIA), *ALK2*, *ALK3*, *ALK6*, *BMP2*, *BMP4*, *BMP6*, *BMP7*, *BMP9*, *IL8*, *NOTCH1*, *NOTCH2*; mouse *Bmp2*, *Bmp6*; and rat *Bmpr2*, *Notch1*, *Notch2*, *Notch3* and *Tnf* Reactions were amplified on a StepOnePlus Real-Time PCR system (Applied Biosystems).

Relative expression of each target gene was identified using the comparative 2-(ΔΔCt) method. Target gene expression was normalized to *ACTB* and the difference in the amount of product produced was expressed as a fold change. The relative abundance of BMP ligands was calculated, on the assumption of equal copy number, by calculating the expression of each BMP gene relative to *ACTB* after normalization to *B2M*.

**Rodent models of PAH.** Group numbers were determined using estimates of variance and minimum detectable differences based on our previous experience with rodent models of PAH. Randomized of animals, using an assigned animal identification number, allowed investigators performing cardiopulmonary measurements to be blinded to genotype or treatment group. Animal studies was conducted in accordance with the UK Animals (Scientific Procedures) Act 1986 and approved under Home Office Project License 80/2460.

**SP-C/*Tnf* mouse model.** Sperm from transgenic SP-C/TNF-α mice (expressing a mouse *Tnf* cDNA driven by the human Surfactant Protein-C promoter) bred on a C56/Bl6 background was kindly provided by Associate Professor Masaki Fujita (Fukuoka University, Japan)[13]. Mice were generated through in vitro fertilization and bred through 2 generations of C57/Bl6 Jax before crossing male SP-C/*Tnf* mice with female *Bmpr2*$^{+/-}$ mice on an established C57/Bl6 Jax background. Offspring were aged to 8–9 weeks and then assessed for pulmonary haemodynamics (Wild-type = 4 females; SP-C/*Tnf* = 4 females; *Bmpr2*$^{+/-}$ = 3 males, 1 female, SP-C/Tnf x *Bmpr2*$^{+/-}$ = 3 males, 1 female).

Mice were anesthetized with 0.5 mg kg$^{-1}$ fentanyl and 25 mg kg$^{-1}$ fluanisone (Hypnorm, VetaPharma Ltd, Leeds, UK) and 12.5 mg kg$^{-1}$ midazolam (Hypnovel), and right ventricular pressures and volumes were recorded using a Millar SPR-139 catheter (Millar Instruments, Houston, TX). Mice were then sacrificed and the hearts, lungs and livers were harvested. Right ventricular hypertrophy (RVH) was assessed by removing the heart and dissecting the right ventricle (RV) free wall from the left ventricle plus septum (LV + S) and weighing separately. The degree of right ventricular hypertrophy was determined from the ratio of RV/LV + S (Fulton Index). The right lung was snap frozen in liquid nitrogen. The left lung was inflated with 0.3% low-melting temperature agarose (Sigma-Aldrich) in phosphate-buffered saline and fixed with 4% paraformaldehyde in PBS before dehydration and paraffin embedding.

**Sugen 5416–hypoxia rat model.** Male Sprague Dawley rats (~150–200 g, Charles River, Saffron Walden, Essex, UK) were administered a single subcutaneous injection of Sugen 5416 (SU-5416; 20 mg kg$^{-1}$, Tocris, Bristol, UK) in vehicle (0.5% carboxyl methylcellulose sodium, 0.4% polysorbate 80, 0.9% benzyl alcohol, all Sigma-Aldrich). Subsequently, rats were placed into a 10% O$_2$ chamber for 3 weeks. After 3 weeks of hypoxia, animals were returned to a normoxic environment for 5 weeks. At 8 weeks, rats were randomized into two groups. One group received i.p. injections of 2.5 mg kg$^{-1}$ Etanercept (Enbrel Pfizer, Sandwich, Kent, UK) diluted in Dulbecco's phosphate-buffered saline (D8537, Sigma-Aldrich) and the second group received vehicle alone. For terminal hemodynamic measurements, rats were anesthetized with xylazine (4.6 mg kg$^{-1}$) and ketamine (7 mg kg$^{-1}$). Body weight was recorded. Right and left ventricular function were assessed using a Millar SPR-869 pressure-volume catheter. Tissue collecting and assessment of right ventricular hypertrophy were conducted as described above for mice.

**Assessment of pulmonary vascular muscularization in mouse and rat tissues.** Paraffin-embedded mouse or rat lung tissue sections (5 µm thick) were incubated with monoclonal mouse-anti-mouse/rat/human smooth muscle α-actin (cat.no. M0851 clone 1A4, Dako; 1:400 dilution). In mouse lung, the primary antibody was detected using the Dako ARK kit (Dako), in accordance with the manufacturer's instructions. Briefly, the primary smooth muscle α-actin antibody was labelled with a biotinylated anti-mouse and then applied to the specimen. A blocking agent, containing mouse serum, was then added to bind residual biotinylation reagent not bound to primary antibody. The biotin-labelled primary antibody was applied to the tissues sections, followed by incubation with

streptavidin-peroxidase. Reaction with diaminobenzidine (DAB)–hydrogen peroxide revealed the cellular location of immunostaining.

Pulmonary arteriolar muscularization was quantified by identifying small pulmonary arteries adjacent to alveolar ducts. Arteries were classified by a blinded observer as non-muscularized, partially muscularized, or fully muscularized, following immunostaining for smooth muscle α-actin. A minimum of 20 vessels (25–75 μm in diameter) were counted from each animal. Statistical significance was determined by comparing the percentage of muscularized vessels between groups.

Percentage wall thickness of muscular small arteries (<100 μm diameter) was quantified in small arteries accompanying terminal bronchioles. Image J was used to measure the diameter of the artery and the thickness of the medial layer following immunostaining for smooth muscle α-actin. To determine wall thickness four measurements were made, one in each quadrant of the artery. A minimum of 10 arteries was measured in each lung section. Percentage wall thickness was calculated as mean wall thickness divided by mean diameter x100. Morphometric measurements were performed in a blinded manner by a single observer, unaware of the experimental groups of the samples.

**NOTCH2 and NOTCH3 immunohistochemistry in lung tissues.** NOTCH2 and NOTCH3 localization and expression in human and rat lung sections was performed by staining fixed tissue sections using rabbit anti-NOTCH2 (cat #ab118824, Abcam, UK; 1:200 dilution) or rabbit anti-NOTCH3 (cat #5276; D11B8, Cell Signaling Technology; 1:800 dilution) and labelled using immunoperoxidase (Vectastain Elite, Vector Laboratories, Peterborough, Cambridgeshire, UK) and 3,3′-DAB to create a brown coloured reaction product.

**TNFα immunofluorescent staining in human lung tissue.** For TNFα immunostaining[63], formalin-fixed paraffin-embedded sections of human lung tissue were subjected to heat mediated antigen retrieval using citrate buffer, pH 6.0 and incubated at 4 °C overnight with 1:50 dilution in blocking buffer of rabbit polyclonal anti-TNFα (IgG; cat no: ab8871 Abcam) and 1:250 dilution of mouse monoclonal anti-human Smooth Muscle Actin (IgG2a, clone: 1A4; cat no: M0851, Dako). After 5 min (×3) washes, sections were further incubated for 1 h at room temperature with 1:100 dilution of secondary antibody in blocking buffer; donkey anti-rabbit Northern Lights IgG-NL493 and donkey anti-mouse-Northern Lights anti-mouse IgG-NL557 (R&D Systems) containing 1 μg ml−1 Hoechst 33342 (cat no: H3570, Thermo Fisher, Loughborough, Leicestershire, UK). Sections were mounted in Vectashield Mounting Media (Vector Laboratories) and imaged with a Leica SPE confocal laser scanning microscope (Leica Microsystems (UK) Ltd, Milton Keynes, UK). Controls included use of isotype-specific primary antibody or non-immune serum.

**Statistics.** All data were analysed using GraphPad Prism and tested for normality using a Kolmogorov-Smirnov test where numbers permitted. Data are presented as mean ± s.e.m. Data were analysed by one-way analysis of variance with *post hoc* Tukey's HSD analysis or paired two-tailed Student's *t*-test where indicated. $P < 0.05$ was considered significant.

**Data availability.** The data that support the findings of this study are available from the corresponding author on reasonable request.

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

## Acknowledgements

This work was supported by grants from the British Heart Foundation, RG/13/4/30107 (N.W.M.), CH/09/001/25945 (N.W.M.), a Medical Research Council Experimental Challenge Award (N.W.M.), a Fondation Leducq Transatlantic Network of Excellence (N.W.M.), the Dinosaur Trust (A.R.) and a UK National Institute for Health Research Healthcare Science Fellowship (M.S.). L.A.H. was funded through a BHF PhD student programme (FS/09/050). The UK National Institute for Health Research Cambridge Biomedical Research Centre and Cell Phenotyping Hub provided infrastructure support.

## Author contributions

L.A.H. designed, performed and analysed the majority of the *in vitro* and some *in vivo* experiments and wrote the manuscript. B.J.D. designed, performed and analysed a significant number of *in vivo* experiments and undertook histological analysis of tissues from *in vivo* experiments and wrote the manuscript. L.L. and A.C. performed and analysed *in vivo* experiments. R.A.L. performed histological staining of human tissues for TNFα. J.D. developed the human BMPR-II ELISA. M.S. stained human and rodent tissues and assisted with analysis. X.D.Y. generated adenovirus and assisted with *in vivo* experiments and tissue harvesting. M.N. performed preliminary TNFα experiments on PAECs. B.H. and G.J.I. generated the C2C12-BRE line. J.R.B. contributed to the writing of the manuscript. A.A.R. contributed to the conception of the study and the design of multiple experiments and contributed to writing the manuscript. P.D.U. conceived the major hypothesis, designed, supervised the majority of experiments and wrote the manuscript. N.W.M. contributed to the conception of the study and the design of multiple experiments and wrote the manuscript.

## Additional information

**Competing financial interests:** The authors declare no competing financial interests.

