## [Peer Review File · Nature Communications]

Reviewers' comments:

Reviewer #1 (expert in PAH)

Remarks to the Author:

This study seeks to address the interaction between TNF α and BMP signalling in the pulmonary vasculature. Using a combination of cell and animal models, the authors report that TNF α promotes ADAM10/17 dependent cleavage of the BMP receptor, BMPR2 (the most commonly mutated in human pulmonary arterial hypertension), in pulmonary vascular smooth muscle cells, leaving the ectodomain to act as a ligand trap. The loss of BMPR2 reduces BMP4 signalling but promotes BMP6 signalling via alternative receptor-mediated pathways, and is associated with alterations in NOTCH signalling and c-SRC. Treatment of a rodent model with an anti-TNF α antibody restored NOTCH signalling and reversed the pulmonary hypertension phenotype.

The studies are well described and the data look internally consistent.

The authors thus identify 3 mechanisms - increased BMP6 signalling, ligand trapping and NOTCH signalling - to explain an interaction between TNF α and BMP.

Increased BMP6 signalling in the presence of reduced BMPR2 expression has been reported before. That TNF α augments BMP6 expression is interesting but as the authors note, it is difficult to reconcile this finding (which would increase Smad signalling) with a proliferative state.

TNF α -induced cleavage of BMPR2, leading to release of the ectodomain for ligand trapping, is new. This shows cell specificity, in that it is only seen in vascular smooth muscle cells, not endothelial cells, and this is related to the cell specific expression of ADAM10/17. But it is seen in aortic as well as pulmonary vascular smooth muscle cells, raising the question of how this might explain why it contributes to pulmonary but not systemic vascular remodelling in vivo.

Where they depart from the published literature is their observations on notch signalling. They observe that TNF α increases NOTCH1 and NOTCH2 but reduces NOTCH3 levels, effects enhanced by BMP6. They argue that NOTCH 1 and 2 are proliferative and NOTCH3 is antiproliferative. This is at odds with the published literature (see reference list below) where levels of NOTCH3 protein are regarded as a sensitive molecular marker of severity of PAH in humans and pulmonary hypertension in rodents. Notch signaling is involved in vascular development and lung tissue from patients with pulmonary hypertension has been reported to show increased NOTCH3 and NOTCH3 intracellular domain expression when compared with normotensive patients. Additionally, NOTCH3 and NOTCH3 intracellular domain expression has been shown to be increased in two animal models of pulmonary hypertension-hypoxia-induced pulmonary hypertension in mice and monocrotaline-induced pulmonary hypertension in rats. The authors reference the difference and suggest that their γ -secretase inhibitor used to inhibit NOTCH cleavage is more selective than the DAPT agent used in published reports but they do not address the fact that NOTCH3 knockout mice are resistant to pulmonary hypertension.

A number of cell lines are used in the study. The eventual focus is on distal pulmonary vascular smooth muscle cells. It is understandable that the investigators focus on one cell type/model, but it does make extrapolation to the whole animal problematic. They have attempted this with mouse models but these do not reflect the human condition. The challenge in extrapolating to humans is evident in the emerging literature where antibodies to DLL4, a NOTCH ligand, in development for the treatment of a range of tumours have been reported to cause pulmonary hypertension in patients.

Minor comments

1. The last sentence in discussion needs rethinking. It is not clear how anti-TNF α can be used "in the prevention of at-risk cohorts".
2. References 35 and 36 are incomplete.

References reporting increased notch3 in pulmonary hypertension

Li X, Zhang X, Leathers R, Makino A, Huang C, Parsa P, Macias J, Yuan JX, Jamieson SW, Thistlethwaite PA. Notch3 signaling promotes the development of pulmonary arterial hypertension. *Nat Med* 2009;15: 1289-1297.

Qiao L1, Xie L, Shi K, Zhou T, Hua Y, Liu H. Notch signaling change in pulmonary vascular remodeling in rats with pulmonary hypertension and its implication for therapeutic intervention. *PLoS One*. 2012;7(12):e51514. doi: 10.1371/journal.pone.0051514. Epub 2012 Dec 12.

Xiao, Y., Gong, D. & Wang, W. Soluble JAGGED1 inhibits pulmonary hypertension by attenuating notch signaling. *Arteriosclerosis, thrombosis, and vascular biology* 33, 2733-2739 (2013).

Yu YR1, Mao L, Piantadosi CA, Gunn MD. CCR2 deficiency, dysregulation of Notch signaling, and spontaneous pulmonary arterial hypertension. *Am J Respir Cell Mol Biol*. 2013 May;48(5):647-54. doi: 10.1165/rcmb.2012-0182OC.

Chida A, Shintani M, Matsushita Y, Sato H, Eitoku T, et al. (2014) Mutations of NOTCH3 in childhood pulmonary arterial hypertension. *Mol Genet Genomic Med* 2: 229-239.

Song Y, Zhang Y, Jiang H, Zhu Y, Liu L, Feng W, Yang L, Wang Y, Li M. Activation of Notch3 promotes pulmonary arterial smooth muscle cells proliferation via Hes1/p27Kip1 signaling pathway. *FEBS Open Bio*. 2015 Aug 12;5:656-60. doi: 10.1016/j.fob.2015.08.007. eCollection 2015.

Zhang Y, Xie X, Zhu Y, Liu L, Feng W, Pan Y, Zhai C, Ke R, Li S, Song Y, Fan Y, Fan F, Wang X, Li F, Li M. Inhibition of Notch3 prevents monocrotaline-induced pulmonary arterial hypertension. *Exp Lung Res*. 2015;41(8):435-43. doi: 10.3109/01902148.2015.1060545. Epub 2015 Aug 28.

Anti-DLL4 antibody and pulmonary hypertension

Chiorean EG, LoRusso P, Strother RM, Diamond JR, Younger A, Messersmith WA, Adriaens L, Liu L, Kao RJ, DiCioccio AT, Kostic A, Leek R, Harris A, Jimeno A. A Phase I First-in-Human Study of Enoticumab (REGN421), a Fully Human Delta-like Ligand 4 (DII4) Monoclonal Antibody in Patients with Advanced Solid Tumors. *Clin Cancer Res*. 2015 Jun 15;21(12):2695-703. doi: 10.1158/1078-0432.CCR-14-2797. Epub 2015 Feb 27.

Reviewer #2 (expert in ADAM proteins)
Remarks to the Author:

This manuscript explores the contribution of a pathway involving TNF α -induced processing of the BMPR2 in the pathogenesis of pulmonary arterial hypertension (PAH). This very thorough study employs a combination of genetic and cell based approaches together with pharmacological inhibitors and siRNA-mediated knockdown of various signaling components to build a quite comprehensive picture of the pathway resulting in pulmonary arterial hypertension. In addition, it provides a compelling explanation for how mutations in the BMPR2 could contribute to the pathogenesis of PAH.

Overall, the data are of high quality, yet there are some concerns regarding the interpretation of the results related to Notch signaling and some additional experimental questions regarding the shedding of the BMPR2 that remain to be addressed.

As to specifics:

1) In the Western blots for ADAM10 and ADAM17 in supplementary figure 5c, it is not clear whether the pro- or mature form of ADAM10 and ADAM17 are shown (please also include molecular weight markers). Cell lysates for Western blots of ADAM17 should be prepared in the presence of a metalloprotease inhibitor such as TAPI-1 and 10 mM 1,10 phenanthroline to block the autodegradation of the mature form, which occurs rapidly following cell lysis in the absence of these inhibitors, and can thus affect the interpretation of the levels of mature ADAM17. It is also important to point out that increased expression of ADAM17 does not necessarily result in increased activity, since this is a post-translationally regulated metalloproteinase (see, for example, PMID:23342154). Nevertheless, the overall interpretation that both ADAMs contribute to shedding of the BMPR2 under the conditions used in this study is well supported by the results.

2) Supplementary Figure 5D presumably needs to be revised so that siADAM17 and D1(A12) are shown next to ADAM17.

3) Regarding the interpretation of data related to Notch signaling, it is important to point out that increased Notch expression does not necessarily lead to increased Notch signaling. Notch signaling is regulated through binding of a membrane-anchored ligand such as Jagged1 or 2, or Dll1, 3 or 4, on a signal-sending cell to a Notch receptor on a signal-receiving cell. The resulting endocytosis of Notch and its ligand is thought to pull open the membrane-proximal negative regulatory domain, allowing access of ADAM10 to the Notch cleavage site. So increased expression of Notch or of ADAM10 will not lead to increased signaling in the absence of this ligand-induced activation of Notch. The two references implicating ADAM17 in Notch signaling are not representative of the majority of papers in this field, in which targeted deletion of ADAM10 in mice typically results in Notch-dependent defects, whereas targeted deletion of ADAM17 does not. Moreover, the Western blots of Notch2 and Notch3 do not show the Notch intracellular domain, but instead show the typical S1-cleaved Notch fragments of around 100 - 120 kD (please include molecular weight markers on all gels). This fragment is constitutively generated by cleavage through furin (or related pro-protein convertases), and is not indicative of signaling (PMID:19379690 provides one of several excellent reviews on this topic). The S2 fragment, generated by ADAM10 under physiological conditions, and the S3 fragment, generated by gamma-secretase, are indicative of ligand-induced Notch processing and signaling, but these are most likely not shown on the various Notch blots presented here. So the authors must make it very clear that these blots only allow conclusions to be drawn about Notch expression levels, not about Notch signaling (the same is true for histochemical data).

Results obtained with DAPT, on the other hand, are indicative of a block of Notch signaling, and can be interpreted in this manner. However, the authors should also discuss that Notch2 and Notch3 might not be active in the same cell types, and in general, offer a more careful interpretation of their data related to Notch expression levels. It is very important to distinguish between differences in Notch expression and Notch signaling, which do not have to be directly related, and to consider different effects of Notch signaling depending on the cell types where it occurs.

Reviewer #3 (expert in PAH and BMP receptors)

Remarks to the Author:

General Comments:

The authors have extensively studied the interaction between BMP and Notch signaling that is dysregulated in response to a specific inflammatory stimulus, TNF-alpha. The cell biology studies have been carefully carried out and the main conclusions indicated in the summary are certainly substantiated and are novel. There are however, other studies in the literature that clearly indicate important points of interaction between the BMPR2 pathway and inflammation not cited, including TNF alpha beginning with Song et al (Circulation, 2005:112; 552-62) but also Hagen et al (Am J Physiol 2007: 292; 1473-9), Lawrie et al, Am J Pathol 2008: 172: 256-64) Sawada et al, 2014: 211; 263-80) Diebold et al, (Cell Metab 2015: 21; 598-608) and many others. Also there are papers on Notch and BMP crosstalk in blood vessels (Rostama et al, ATVB 2015: 35; 2626-37). The authors also do not reconcile their results with their recent studies related to rescue of PAH by the ligand BMP9, in fact surprisingly, work with BMP9 as a ligand seems to be completely excluded. The SUGEN/hypoxia model treated with the TNF soluble receptor reverses or retards progression of the pressure and right ventricular hypertrophy in concert with elevating Notch signaling but the impact on remodeling is less severe and the role of TNF alpha (despite its elevation) vis a vis other cytokines is not clear.

Specific Comments:

Introduction:

Page 3 last paragraph, see above comments. The impact of TNFalpha and loss of BMPR2 was studied in Sawada et al cited above, although the paper did not show that TNF specifically reduces BMPR2 levels in endothelial cells.

Page 5 first paragraph: The mechanism by which TNF reduces BMPR2 in endothelial cells is never addressed.

Page 6, second paragraph. It is hard to visualize why two disintegrins are necessary to cleave the same site. This needs further explanation.

Page 6, third paragraph. In view of the authors' recent work showing a protective effect of BMP9, it would be interesting to determine whether BMP9 signaling is compromised by TNF alpha. (Also true for data presented on Page 7).

Page 10: The controversy with the Li et al paper cited is still problematic. Notch3 and Notch3 ICD are elevated in PAH PA SMC and in human PAs in tissue sections. The blots in Figure 4 should definitely be quantified because it appears that Notch 3 mRNA is higher in the PAH vs. control PA SMC. It is hard to make much of differences in immunoblots and immunohistochemistry because they rely on the affinity of the relative antibodies. The authors should be sure that the siRNAs they use are specific, ie. Do Notch 3 and Notch 2 reduce with Notch 2 siRNA? Does Notch 3 but not Notch 2 decrease with siRNA for Notch 3. Those data should be included. Finally it does appear from the mRNA data in the supplement that the effect of Smad6 induced by TNFalpha is different, increasing Notch 2 and reducing Notch 3. Thus the TNF effect may be selective to Notch 3 particularly in the setting of loss of BMPR2. In the mice, knockout of Notch 3 is protective, so there is more to try to reconcile either in the text or in the discussion. It should also be noted that the PASMCM come from familial PAH with a BMPR2 mutation so here again the populations are different.

Page 11: The selective effects of the FYN and YES siRNAs should be documented in view of the results.

Interestingly the TNF overexpression does not result in more severe RVSP or RVH when BMPR2 is heterozygous. On the other hand, in the SUGEN/Hypoxia model there is clear suppression of RVSP and RVH by the TNF soluble receptor but less impact on the muscularization of vessels. The authors should discuss these differences.

Discussion

Page 13: The discussion with respect to Notch seems to short-change the previous observations,

particularly related to the prevention of pulmonary hypertension in the Notch 3 knockout mice. There is no question, however, that the mechanistic studies in this paper are comprehensive and well thought through.

Reviewer #4 (expert in PAH and BMP receptors)

Remarks to the Author:

The manuscript describes an interesting and complex system involving a switch in TGF β receptor super-family usage in PSMCs from an ALK3 / BMPR-II-driven system to an ACTR-IIA / ALK2 system-driven primarily by TNF α degradation of BMPR-II and the induction in expression of various genes including BMP6 and ACTR-IIA. In the background of compromised BMPR-II expression / function as might be found in PSMCs from patients with heritable forms of Pulmonary Arterial Hypertension (PAH) and genetic mouse models of PAH, this alteration in receptor usage results in a switch from an anti-proliferative response of these cells to a pro-proliferative response following stimulation with BMP6. To add to the complexity, TNF α -mediated modulation in the expression of components of the NOTCH pathway with induction and repression of NOTCH2 and NOTCH3 respectively, directly contributes to the pro-proliferative effects of PSMCs in a Src-kinase family-dependent manner. Although the mechanism described in the manuscript is novel and provides several new therapeutic options for the treatment of heritable PAH, I do have some questions that need to be addressed and suggestions for improvement of the manuscript outlined below before it should be considered for publication.

1. The authors present data showing that TNF α induces degradation of BMPR-II via an ADAM10/17-mediated mechanism that appears to be operative in PSMCs but not in PAECs. The expression of BMPR-II is clearly suppressed in PAECs in response to TNF α stimulation, however the authors do not provide any explanation as to the mechanism in this cell type. Do the authors have any mechanistic insights into how BMPR-II protein levels are regulated in PAECs via TNF α stimulation that could be included in the manuscript?

2. The authors have used the SP-C/TNF transgenic mouse model to demonstrate that BMPR-II expression in the whole lung of these animals also appears to be under the control of the TNF α pathway. The data presented by the authors seems to imply that the expression of BMPR-II in whole lung homogenates is almost completely suppressed in this model. This finding seems incongruent with the alveolar epithelial cell-restricted expression of TNF α previously described in this model (Miyazaki et al. 1995 J Clin Invest. 1995 Jul; 96(1):250-9). Immunohistological analysis of BMPR-II and TNF α expression in the lungs of these animals would provide additional spatial information confirming that loss of BMPR-II expression is co-localised to areas of TNF α expression.

3. The legend for Figure 1e needs to have a better description of what stains / proteins the colours in the images refer to. From the images presented, it also appears that α SMA positive cells appear to express TNF α which might suggest that PSMCs isolated from patients with heritable PAH may express higher baseline levels of TNF α in culture. Have the authors determined whether this is the case?

4. The authors have cited information showing that the BMPR-II cytoplasmic tail can act as a scaffolding element that can bind to and modulate the activity of a number of kinases including c-Src (Wong et al. Am J Respir Cell Mol Biol. 2005 Nov; 33(5):438-46). Can I presume that the generation of the BMPR-II ICD might still be capable of binding to kinases like c-Src and modulating their activity? If so, it would then be important to know how long the BMPR-II ICD remains within the cytoplasm.

5. Previously published information by the authors has shown that PSMCs isolated from patients with PAH lose their growth suppressive responses to BMP ligands including BMP2 and BMP4 (Morrell et al. Circulation. 2001 Aug 14; 104(7):790-5), which is in contrast to the data depicted in Figure 2D. Could the authors provide an explanation for the differences observed in this study compared to the previously published report?

6. There are issues with Figure 4 and supplementary Figures 11 and 14:

a) Presumably Figure 4a, 4b and Supplementary Figure 11a, 14d depict NOTCH1, NOTCH2, and NOTCH3 ICD and not full-length protein as indicated in the main body of the manuscript. The Figure legends need to be clarified that what is being depicted are indeed the ICD's and not full-length NOTCH proteins.

b) On page 9 of the manuscript the authors state that in Figure 4a and Supplementary Figure 14c 'BMP2 silencing also promoted the TNF α -dependent reduction of NOTCH3-ICD generation and NOTCH3 transcription'. On inspection of Figure 4a, TNF α does induce a reduction in NOTCH3 ICD levels; however, BMP2 siRNA does not enhance the effect of TNF α on NOTCH3 ICD levels compared to the siCP or DH1 control. Similarly, in Supplementary Figure 14c, TNF α does reduce the transcription of NOTCH3 mRNA; however, this effect is not enhanced by BMP2 siRNA compared to the siCP or DH1 controls.

c) Also on page 9 of the manuscript the authors state in Figure 4b 'In BMP2 heterozygous HPAH PSMCs treated with TNF α , siACVR2A reduced NOTCH2-ICD generation and abrogated NOTCH3-ICD'. On inspection of Figure 4b, treatment of HPAH PSMCs with siACVR2A seems to be without effect on TNF α -mediated NOTCH2-ICD generation.

d) In Figure 4b, treatment of HPAH PSMCs stimulated with 0.1% serum with DH1 alone seems to have had a significant suppressive effect on NOTCH1 protein expression. This does not appear to be due to a loading issue as the authors state that the gels were re-probed for α -tubulin which appears to be uniformly expressed in all lanes on the gel. As this gel is representative of the other replicates, is the effect of DH1 on NOTCH1 expression also observed in the other replicates?

7. The authors have utilised the γ -secretase inhibitor, DAPT in experiments to support the notion that NOTCH ICD generation is involved in the TNF α and BMP6-induced proliferative responses of PSMCs. The concentration of DAPT used in this study was 5mM. Are the authors confident that other proteases important in their proposed mechanism are unaffected by this concentration of DAPT? How does 5mM DAPT affect TNF α -mediated generation of BMPR-II ICD?

8. In Figure 4e, the authors state that expression of NOTCH2 in the medial layer of the pulmonary arteriolar lesion depicted in the lung section taken from an HPAH patient is enhanced compared to control vessels. The intensity of the staining looks somewhat similar between the normal vessel and HPAH vessel in the sections depicted.

9. In Supplementary Figure 17b, Etanercept did not change the % wall thickness for all vessels yet in Figure 6c there appears to be a modest effect of Etanercept on the % of muscularized vessels with Etanercept causing a small increase in non-muscularized vessels compared to the S/H control. Are these endpoints essentially measuring the same thing i.e. pulmonary vascular wall remodelling and, if so, is the data inconsistent?

10. In the body of the text, the authors state repeatedly that the transient induction of Smad1/5 responses by BMP6 / ACTR-IIA / ALK2 signalling is unlikely to promote heightened PSMC proliferation. There are no data shown in the manuscript or supplementary materials that indicate whether the Smad1/5 responses mediated by BMP6 / ACTR-IIA / ALK2 are transient or otherwise. If the authors cannot qualify this statement with data or a citation, then the authors should desist from describing the responses as transient.

Thank you for the thorough and thoughtful reviews of our manuscript. The reviewers were consistent in their recognition of the high quality and internal consistency of our findings and the novelty in several areas that provide new insights into the pathobiology of PAH and offer tractable targets for therapeutic intervention, including repurposing of anti-TNF α approaches.

Our specific responses to the reviewers' comments are below:

Reviewer #1 (expert in PAH)

Remarks to the Author:

Comment 1: This study seeks to address the interaction between TNF α and BMP signalling in the pulmonary vasculature. Using a combination of cell and animal models, the authors report that TNF α promotes ADAM10/17 dependent cleavage of the BMP receptor, BMPR2 (the most commonly mutated in human pulmonary arterial hypertension), in pulmonary vascular smooth muscle cells, leaving the ectodomain to act as a ligand trap. The loss of BMPR2 reduces BMP4 signalling but promotes BMP6 signalling via alternative receptor-mediated pathways, and is associated with alterations in NOTCH signalling and c-SRC. Treatment of a rodent model with an anti-TNF α antibody restored NOTCH signalling and reversed the pulmonary hypertension phenotype. The studies are well described and the data look internally consistent. The authors thus identify 3 mechanisms - increased BMP6 signalling, ligand trapping and NOTCH signalling - to explain an interaction between TNF α and BMP.

Response 1: *We thank the reviewer for recognising the high quality of these data and the new mechanisms revealed by our studies.*

Comment 2: Increased BMP6 signalling in the presence of reduced BMPR2 expression has been reported before. That TNF α augments BMP6 expression is interesting but as the authors note, it is difficult to reconcile this finding (which would increase Smad signalling) with a proproliferative state.

Response 2: *The reviewer is correct that increased BMP6 expression, coupled with a reduction in smooth muscle cells BMPR-II, has been reported by our lab and others to lead to an increase in p-Smad1/5 signalling via ActR-IIa. However, our lab and others have consistently shown that the Smad1/5 signalling pathway is anti-proliferative in PSMCs (Yang et al Circ Res 2005, etc). In addition, the increased Smad signalling via ActR-IIa is reported to be transient and therefore unlikely to be responsible for the sustained proliferative response to BMP6 that we have observed. To confirm*

the transient nature of this Smad response we have included new experiments demonstrating that BMPR-II knockdown leads to increased p-Smad1/5 activity in response to BMP6 (Supplementary Fig. 7b,c) but that this response is very limited in duration (<4 hours). We have added text referring to this observation on page 7.

In contrast to this transient response, the sustained signalling pathway identified by our studies as responsible for hyper-proliferation of PASMCs is robustly shown to be aberrant Notch signalling downstream of ALK2/ActR1a.

Comment 3. TNF α -induced cleavage of BMPR2, leading to release of the ectodomain for ligand trapping, is new. This shows cell specificity, in that it is only seen in vascular smooth muscle cells, not endothelial cells, and this is related to the cell specific expression of ADAM10/17. But it is seen in aortic as well as pulmonary vascular smooth muscle cells, raising the question of how this might explain why it contributes to pulmonary but not systemic vascular remodelling in vivo.

Response 3. *As the reviewer points out, the smooth muscle cell specific cleavage of BMPR-II we report is novel. Indeed, both pulmonary artery smooth muscle cells and human aortic smooth muscle cells exhibit sBMPR-II shedding in response to TNF α . It is entirely reasonable for the reviewer to propose that TNF α -mediated suppression of BMPR-II in the systemic circulation might also drive vascular remodelling. We did not address this in systemic vascular injury models. However, the models employed in our manuscript are specific to pulmonary arterial hypertension, where there is local production of TNF α by inflammatory cells and, as shown in Figure 1e, in the medial layer of the pulmonary artery. Since pulmonary hypertension is a disease state specific to the lung vasculature, there is no major impact on systemic vessels.*

Comment 4. Where they depart from the published literature is their observations on notch signalling. They observe that TNF α increases NOTCH1 and NOTCH2 but reduces NOTCH3 levels, effects enhanced by BMP6. They argue that NOTCH 1 and 2 are proproliferative and NOTCH3 is antiproliferative. This is at odds with the published literature (see reference list below) where levels of NOTCH3 protein are regarded as a sensitive molecular marker of severity of PAH in humans and pulmonary hypertension in rodents. Notch signaling is involved in vascular development and lung tissue from patients with pulmonary hypertension has been reported to show increased NOTCH3 and NOTCH3 intracellular domain expression when compared with normotensive patients. Additionally, NOTCH3 and NOTCH3 intracellular domain expression has been shown to be increased in two animal models of pulmonary hypertension-hypoxia-induced pulmonary hypertension in mice and monocrotoline-induced pulmonary hypertension in rats. The authors reference the difference and suggest that their γ -secretase inhibitor used to inhibit NOTCH cleavage is more selective than the DAPT agent used in published reports but they do not address the fact that NOTCH3 knockout mice are resistant to pulmonary hypertension.

Response 4. *The reviewer highlights that our observations on NOTCH appear to be at odds with some of the published literature in the pulmonary hypertension field. Indeed, when we embarked on these experiments we expected to see the previously reported increase in NOTCH3 and Hes signalling, originally demonstrated in the manuscript by Li et al (Nat Med 2009). Unexpectedly, we consistently observed suppression of NOTCH3 in our model systems and increased expression and signalling via NOTCH2 and HEY1/2, both in vivo and in vitro. We went on to confirm that the NOTCH2 pathway was*

involved in PASMC proliferation and that NOTCH3 appeared anti-proliferative. In fact, we used the same γ -secretase inhibitor, DAPT, as used in previous reports. However, since DAPT has no selectivity for individual NOTCH pathways we also employed siRNA knockdown of NOTCH2 and HEY1/2 to confirm the proproliferative function of this pathway in PASMCs. Our data are internally consistent in multiple experiments in vitro and in vivo.

The reviewer provides a list of previous publications that at first sight appear at odds with our findings. Here we critically appraise these manuscripts in order to, where possible, provide the best explanation for apparent differences. The totality of previously published papers is that there are significant inconsistencies and/or deficiencies in many of these manuscripts that make direct comparison with our data difficult. Most failed to assess NOTCH2 levels. Our studies mainly concern PAH in which BMPR-II expression is reduced markedly and TNF α is activated. This is of less relevance to the study of pure hypoxia-induced PAH in which the downregulation of BMPR-II is minimal. Nevertheless, our data both in vitro and in vivo are entirely consistent between models.

References reporting increased notch3 in pulmonary hypertension

Li X, Zhang X, Leathers R, Makino A, Huang C, Parsa P, Macias J, Yuan JX, Jamieson SW, Thistlethwaite PA. Notch3 signaling promotes the development of pulmonary arterial hypertension. *Nat Med* 2009;15: 1289-1297.

*The analysis of NOTCH signalling in this manuscript was confined to NOTCH3 and did not assess contributions from other NOTCHs. The authors used an antibody from Santa Cruz for detection of the NOTCH3 ICD (catalog number: sc-7424). We tested this antibody in cell lysates, and despite extensive experience with immunoblotting, we found that this antibody performed very poorly compared with the antibodies we sourced from Cell Signaling Technologies. We confirmed the identity of the bands detected by NOTCH antibodies used in our experiments by siRNA knockdown (Supplementary Fig. 13g). One of the central observations in the Li et al. manuscript, as mentioned by the reviewer, was that NOTCH3 knockout mice are protected from hypoxia-induced pulmonary hypertension. We would strongly contest whether this observation supports a specific role for NOTCH3 in the development of pulmonary hypertension, since Notch3 knockout mice demonstrate grossly abnormal arterial maturation in all vascular beds with altered myogenic responses and structural defects (Domenga et al. *Genes Dev* 2004). In view of this, it is not surprising that a mouse deficient in Notch3 during development exhibits a deficient response to chronic hypoxic exposure, as well as any other contractile or remodelling stimulus. As mentioned above, the therapeutic benefits of DAPT used in this manuscript could equally well implicate a role for any of the NOTCHs in vascular remodelling.*

Qiao L1, Xie L, Shi K, Zhou T, Hua Y, Liu H. Notch signaling change in pulmonary vascular remodeling in rats with pulmonary hypertension and its implication for therapeutic intervention. *PLoS One*. 2012;7(12):e51514. doi: 10.1371/journal.pone.0051514. Epub 2012 Dec 12.

This manuscript attempted to more fully characterise the changes in NOTCH1-4 expression in the rat lung during development of hypoxia-induced pulmonary hypertension. These authors did not use immunoblotting for NOTCHs, which is concerning, but instead relied on changes in mRNA expression over 4 weeks. These authors did not detect any changes in NOTCH2 transcripts, but did demonstrate

a transient increase in NOTCH3 at 1 week of hypoxia, which returned to baseline levels by 2 weeks. This observation is also at odds with the manuscript by Li et al, who reported sustained increases in NOTCH3 in hypoxic mice. Interestingly, and confusingly, Qiao et al. reported an increase in HERP1 (aka HEY2) during chronic hypoxia, which we have confirmed is a target gene of NOTCH2.

Xiao, Y., Gong, D. & Wang, W. Soluble JAGGED1 inhibits pulmonary hypertension by attenuating notch signaling. *Arteriosclerosis, thrombosis, and vascular biology* 33, 2733-2739 (2013).

This manuscript did not examine changes in NOTCH2 expression. The authors did not directly assess the expression of NOTCH2 and NOTCH3 in lung tissue from monocrotaline rats and hypoxic mice. Instead they isolated PSMCs from these animals and grew them in culture for an unspecified time before isolating and measuring NOTCH1 and NOTCH3 mRNA expression. Jagged1 levels were measured in whole lung and were correlated with mean pulmonary arterial pressure. The ascertainment of NOTCH2 and 3 levels in isolated PSMCs are very different to the approach used in our experiments (which used whole lung) and it is not clear why the authors used a different approach since NOTCH levels will be greatly influenced by culture in serum and multiple passages. The main emphasis of this manuscript by Xiao et al. is that Jagged1 is increased in the PAH models and that soluble Jagged1 inhibits PAH. These results are consistent with our findings since Jagged1 is likely the important NOTCH ligand and can signal via NOTCH2 and/or NOTCH3.

Yu YR1, Mao L, Piantadosi CA, Gunn MD. CCR2 deficiency, dysregulation of Notch signaling, and spontaneous pulmonary arterial hypertension. *Am J Respir Cell Mol Biol.* 2013 May;48(5):647-54. doi: 10.1165/rcmb.2012-0182OC.

These authors assessed the transcript levels of lung NOTCH1-4 during chronic hypoxic exposure (up to 4 weeks) and observed significantly increased NOTCH3 mRNA expression only after 4 weeks of hypoxia. Protein levels of full length or intracellular domains of NOTCH were not assessed in this study. Interestingly the authors observed no increase in HES-5 expression in hypoxic lung of control mice, which is the specific target of NOTCH3 signalling, thus providing no evidence for increased NOTCH3 signalling during chronic hypoxia in the mouse.

Chida A, Shintani M, Matsushita Y, Sato H, Eitoku T, et al. (2014) Mutations of NOTCH3 in childhood pulmonary arterial hypertension. *Mol Genet Genomic Med* 2: 229-239.

This report of loss-of-function NOTCH3 mutations that lead to reduced NOTCH3 signalling and cell hyperproliferation in children with PAH is at odds with the findings of Li et al. Nat Med 2009, but entirely compatible with our results.

Song Y, Zhang Y, Jiang H, Zhu Y, Liu L, Feng W, Yang L, Wang Y, Li M. Activation of Notch3 promotes pulmonary arterial smooth muscle cells proliferation via Hes1/p27Kip1 signaling pathway. *FEBS Open Bio.* 2015 Aug 12;5:656-60. doi: 10.1016/j.fob.2015.08.007. eCollection 2015.

These authors showed that forced overexpression of the NOTCH3 intracellular domain by adenoviral transfection drives proliferation of rat PSMCs in culture via HES1. In this paper, contrary to the

findings of Li et al. Nat Med 2009, rat PSMCs expressed little or no NOTCH3 at baseline. Expression of NOTCH2 was not assessed. It is difficult to compare the results of this overexpression study with the examination of endogenous signalling pathways in the context of BMP signalling, TNF α and reduced BMPR-II expression as undertaken in our experiments.

Zhang Y , Xie X, Zhu Y, Liu L, Feng W, Pan Y, Zhai C, Ke R, Li S, Song Y, Fan Y, Fan F, Wang X, Li F, Li M. Inhibition of Notch3 prevents monocrotaline-induced pulmonary arterial hypertension. Exp Lung Res. 2015;41(8):435-43. doi: 10.3109/01902148.2015.1060545. Epub 2015 Aug 28.

These authors showed increased levels of NOTCH3 ICD in the lungs of rats with monocrotaline-induced PAH. NOTCH2 was not assessed. The use of DAPT as an inhibitor of NOTCH in these studies cannot differentiate between the effects of any NOTCH isoform.

Anti-DLL4 antibody and pulmonary hypertension

Chiorean EG, LoRusso P, Strother RM, Diamond JR, Younger A, Messersmith WA, Adriaens L, Liu L, Kao RJ, DiCioccio AT, Kostic A, Leek R, Harris A, Jimeno A. A Phase I First-in-Human Study of Enoticumab (REGN421), a Fully Human Delta-like Ligand 4 (DLL4) Monoclonal Antibody in Patients with Advanced Solid Tumors. Clin Cancer Res. 2015 Jun 15;21(12):2695-703. doi: 10.1158/1078-0432.CCR-14-2797. Epub 2015 Feb 27.

This Phase I study, and others, have reported a significant incidence of pulmonary hypertension in patients exposed to anti-DLL4 antibody treatments for cancer. The pulmonary hypertension is usually reversible. This side effect is thought to be due to inhibition of NOTCH signalling in endothelial cells, particularly in the tip cells of angiogenic sprouts, which enhances VEGF signalling, so most likely functions through a different mechanism to the smooth muscle responses we examine in this manuscript. The effect of anti-DLL4 also raises concerns regarding therapeutic approaches that broadly inhibit NOTCH signalling such as DAPT. More specific approaches targeting smooth muscle cell proliferation, such as anti-jagged1, may be less likely to adversely promote the development of PAH.

Comment 5. A number of cell lines are used in the study. The eventual focus is on distal pulmonary vascular smooth muscle cells. It is understandable that the investigators focus on one cell type/model, but it does make extrapolation to the whole animal problematic. They have attempted this with mouse models but these do not reflect the human condition. The challenge in extrapolating to humans is evident in the emerging literature where antibodies to DLL4, a NOTCH ligand, in development for the treatment of a range of tumours have been reported to cause pulmonary hypertension in patients.

Response 5. *We have included data derived from human pulmonary artery smooth muscle cells and endothelial cells, in addition to cell lines. In addition, we have included relevant and informative animal models to show how this pathway is regulated in vivo. Taken together, the data are entirely internally consistent from human cells to animal models. The emerging literature on a causal role for DLL4 inhibition in PAH is not at odds with our findings since the likely pathway implicated in these anti-DLL4 studies is inhibition of NOTCH signalling in endothelial cells. As pointed out by the reviewer,*

the anti-TNF α approach, central to our study, leads to inhibition of dysregulated NOTCH signalling in pulmonary artery smooth muscle cells.

Minor comments

Comment 1. The last sentence in discussion needs rethinking. It is not clear how anti-TNF α can be used "in the prevention of at-risk cohorts".

Response 1: *This comment has been removed from the Discussion and we have clarified our patient target group on page 15.*

Comment 2. References 35 and 36 are incomplete.

Response 2: *Thank you for highlighting this omission, we have updated the references to include the PLOSOne e-page numbers.*

Reviewer #2 (expert in ADAM proteins)

Remarks to the Author:

This manuscript explores the contribution of a pathway involving TNF α -induced processing of the BMPR2 in the pathogenesis of pulmonary arterial hypertension (PAH). This very thorough study employs a combination of genetic and cell based approaches together with pharmacological inhibitors and siRNA-mediated knockdown of various signaling components to build a quite comprehensive picture of the pathway resulting in pulmonary arterial hypertension. In addition, it provides a compelling explanation for how mutations in the BMPR2 could contribute to the pathogenesis of PAH. Overall, the data are of high quality, yet there are some concerns regarding the interpretation of the results related to Notch signaling and some additional experimental questions regarding the shedding of the BMPR2 that remain to be addressed.

As to specifics:

Comment 1: In the Western blots for ADAM10 and ADAM17 in supplementary figure 5c, it is not clear whether the pro- or mature form of ADAM10 and ADAM17 are shown (please also include molecular weight markers). Cell lysates for Western blots of ADAM17 should be prepared in the presence of a metalloprotease inhibitor such as TAPI-1 and 10 mM 1,10 phenanthroline to block the autodegradation of the mature form, which occurs rapidly following cell lysis in the absence of these inhibitors, and can thus affect the interpretation of the levels of mature ADAM17. It is also important to point out that increased expression of ADAM17 does not necessarily result in increased activity, since this is a post-translationally regulated metalloproteinase (see, for example, PMID:23342154). Nevertheless, the overall interpretation that both ADAMs contribute to shedding of the BMPR2 under the conditions used in this study is well supported by the results.

Response 1: We are thankful for the reviewer's insightful comments regarding the autodegradation of ADAM10 and ADAM17 and the requirement for the addition of inhibitors to block this. The study detailing this autodegradation (Schlöndorff J, Becherer JD, Blobel CP. 2000. Intracellular maturation and localization of the tumour necrosis factor alpha convertase (TACE). *Biochem. J.* 347(Part 1):131–138) examined the process in lysis buffers containing EDTA and Triton. We homogenised our frozen tissue with a more aggressive lysis buffer containing 2% SDS, which we might expect to denature enzyme activity and reduce the potential for autodegradation. We acknowledge that this may not be entirely sufficient, but we did not observe any 60kDa band observed representing the autodegraded form.

For confirmation of the siRNA knockdown in vascular cells in Fig. 1h, we observe definitive losses of ADAM proteins only with their specific siRNAs. The use of appropriate inhibitors is an important detail, but as the reviewer points out, the overall interpretation that both ADAMs contribute to BMPR2 shedding is well supported by the experimental data provided. We have included a sentence on page 6 of the results section stating that the ADAM levels alone may not be reflective of altered activity. As requested by the reviewer we have now included molecular weight markers for ADAM10 and 17 on the immunoblots (Fig. 1h, Supplementary Fig. 5c).

Comment 2: Supplementary Figure 5D presumably needs to be revised so that siADAM17 and D1(A12) are shown next to ADAM17.

Response 2: The reviewer is correct in their observation that siADAM17 and D1(A12) should be shown next to ADAM17 in Supplementary Fig. 5d. We apologise for this oversight and have corrected the figure.

Comment 3: Regarding the interpretation of data related to Notch signaling, it is important to point out that increased Notch expression does not necessarily lead to increased Notch signaling. Notch signaling is regulated through binding of a membrane-anchored ligand such as Jagged1 or 2, or DLL1, 3 or 4, on a signal-sending cell to a Notch receptor on a signal-receiving cell. The resulting endocytosis of Notch and its ligand is thought to pull open the membrane-proximal negative regulatory domain, allowing access of ADAM10 to the Notch cleavage site. So increased expression of Notch or of ADAM10 will not lead to increased signaling in the absence of this ligand-induced activation of Notch. The two references implicating ADAM17 in Notch signaling are not representative of the majority of papers in this field, in which targeted deletion of ADAM10 in mice typically results in Notch-dependent defects, whereas targeted deletion of ADAM17 does not. Moreover, the Western blots of Notch2 and Notch3 do not show the Notch intracellular domain, but instead show the typical S1-cleaved Notch fragments of around 100 - 120 kD (please include molecular weight markers on all gels). This fragment is constitutively generated by cleavage through furin (or related pro-protein convertases), and is not indicative of signaling (PMID:19379690 provides one of several excellent reviews on this topic). The S2 fragment, generated by ADAM10 under physiological conditions, and the S3 fragment, generated by gamma-secretase, are indicative of ligand-induced Notch processing and signaling, but these are most likely not shown on the various Notch blots presented here. So the authors must make it very clear that these blots only allow conclusions to be drawn about Notch expression levels, not about Notch signaling (the same is true for histochemical data).

Response 3: *The reviewer is correct in their statement that we are most likely detecting the S1 cleaved products of NOTCH1, NOTCH2 and NOTCH 3 and we have stated this on page 9 and the figure legends for the figures listed below. We have added the positions of the closest protein standard to the NOTCH bands in PSMCs in Fig. 4a, Fig. 4b, Supplementary Fig. 11c, Supplementary Fig. 13g, Supplementary Fig. 14f and the newly added Supplementary Fig. 11a, 14d and 14e. In summary, NOTCH1 was just over 120kDa, NOTCH2 was 100kDa and NOTCH3 was a doublet between 90-95kDa. We do not claim any specific selectivity of ADAM10 or ADAM17 for the cleavage but, as the reviewer suggests, we have now included more representative citations implicating ADAM10 in this process. We now clarify on page 9 that Notch expression levels do not reflect signalling, but that the data regarding HES/HEY responses indicate signalling.*

Comment 4: Results obtained with DAPT, on the other hand, are indicative of a block of Notch signaling, and can be interpreted in this manner. However, the authors should also discuss that Notch2 and Notch3 might not be active in the same cell types, and in general, offer a more careful interpretation of their data related to Notch expression levels. It is very important to distinguish between differences in Notch expression and Notch signaling, which do not have to be directly related, and to consider different effects of Notch signaling depending on the cell types where it occurs.

Response 4: *We have gone through the original description of our findings and made sure that we give due consideration to these points. We have specified that we are examining the NOTCH cleaved/transmembrane intracellular (NTM) regions in the results section (page 9) and have only referred specifically to NOTCH signalling in the context of the HEY/HES transcriptional responses and the impact of HEY1 and HEY2 siRNAs on PSMC proliferation. In particular we now include consideration in the Discussion (pages15/16) that the Notch signalling needs to be considered in the cellular context.*

Reviewer #3 (expert in PAH and BMP receptors).

Remarks to the Author:

General Comments:

Comment 1: The authors have extensively studied the interaction between BMP and Notch signaling that is dysregulated in response to a specific inflammatory stimulus, TNF-alpha. The cell biology studies have been carefully carried out and the main conclusions indicated in the summary are certainly substantiated and are novel. There are however, other studies in the literature that clearly indicate important points of interaction between the BMPR2 pathway and inflammation not cited, including TNF alpha beginning with Song et al (Circulation, 2005:112; 552-62) but also Hagen et al (Am J Physiol 2007: 292; 1473-9), Lawrie et al, Am J Pathol 2008: 172: 256-64) Sawada et al, 2014: 211; 263-80) Diebold et al, (Cell Metab 2015: 21; 598-608) and many others. Also there are papers on Notch and BMP crosstalk in blood vessels (Rostama et al, ATVB 2015: 35; 2626-37). The authors also do not reconcile their results with their recent studies related to rescue of PAH by the ligand BMP9, in fact surprisingly, work with BMP9 as a ligand seems to be completely excluded. The SUGEN/hypoxia model treated with the TNF soluble receptor reverses or retards progression of the pressure and right ventricular hypertrophy in concert with elevating Notch signaling but the impact

on remodeling is less severe and the role of TNF alpha (despite its elevation) vis a vis other cytokines is not clear.

Response 1: We thank the reviewer for their very positive comments regarding our manuscript including the novelty. We recognise the contribution of others in the field who have shown that inflammation is an important aspect of PAH and BMPR2 signalling, although none have dissected the mechanistic pathways to the extent presented in our manuscript. We thank the reviewer for the suggestion of further references that could be included. We have included the reference relating to the promotion of PAH in *Bmpr2*^{+/-} mice (Song et al. 2005) in the introduction (page 3 and Reference 10). The references by Hagen et al (IL6 and PAH in mice) and Lawrie et al (osteoprotegerin /*TNFRSF11B* increased in PAH) are not directly related to the theme of this manuscript. We have not included the reference by Rostama et al (DLL4 and BMP9 relating to EC survival and Thrombospondin 1) as it is not directly related to the theme, but have included a paragraph in the discussion (pages15/16) to include a reference to endothelial cells.

We have undertaken extensive studies of BMP9 signalling in endothelial cells and the impact of TNF α on this response. As expected, TNF α inhibits BMP9 signalling in a BMPR2-specific manner. However, these studies are central to another manuscript that is in preparation that specifically characterises the mechanisms of BMPR2 signalling by BMP9 in ECs. Given the length and complexity of the current manuscript it would be difficult to combine these data. We have now included a sentence that such studies are warranted and underway.

Specific Comments:

Introduction:

Comment 2: Page 3 last paragraph, see above comments. The impact of TNFalpha and loss of BMPR2 was studied in Sawada et al cited above, although the paper did not show that TNF specifically reduces BMPR2 levels in endothelial cells.

Response 2: The reviewer is correct that the impact of TNF α in the setting of reduced BMPR2 levels has been studied in the manuscript by Sawada et al, 2014. Indeed it is interesting that these authors identified other mechanisms by which TNF α exerts exaggerated effects when BMPR2 levels are reduced (in that manuscript via increased GM-CSF). We have now cited this manuscript in the revised Discussion (pages 15/16 and reference 53).

Comment 3: Page 5 first paragraph: The mechanism by which TNF reduces BMPR2 in endothelial cells is never addressed.

Response 3: We have now included new experimental data relating to PAECs in Supplementary Figure 1e. We have shown that the TNF α -dependent repression of BMPR-II is mediated by p65(RelA) and refer to this on page 4.

Comment 4: Page 6, second paragraph. It is hard to visualize why two disintegrins are necessary to cleave the same site. This needs further explanation.

Response 4: *Our knockdown results clearly show that each of these ADAMs can compensate for the loss of the other. Thus only dual knockdown or inhibition inhibits TNF α -mediated cleavage of BMPR-II but we are unsure of the reason for the redundancy. It is possible that ADAM10 and ADAM17 may be required to be incorporated into a complex that is necessary for BMPR-II cleavage, to ensure that this event can only take place in cells where both enzymes are expressed. This has now been clarified in the first paragraph of the discussion (page 13).*

Comment 5: Page 6, third paragraph. In view of the authors' recent work showing a protective effect of BMP9, it would be interesting to determine whether BMP9 signaling is compromised by TNF alpha. (Also true for data presented on Page 7).

Response 5: *As mentioned above, we have now undertaken extensive studies of BMP9 signalling in endothelial cells and the impact of TNF α on this response. As expected, TNF α inhibits BMP9 signalling in a BMPR2-specific manner. However, these studies are central to another manuscript that is in preparation that specifically characterises the mechanisms of BMPR2 signalling by BMP9 in ECs. Given the length and complexity of the current manuscript it would be difficult to combine these data. We have now included a sentence that such studies are warranted and underway (page 16).*

Comment 6: Page 10: The controversy with the Li et al paper cited is still problematic. Notch3 and Notch3 ICD are elevated in PAH PA SMC and in human PAs in tissue sections. The blots in Figure 4 should definitely be quantified because it appears that Notch 3 mRNA is higher in the PAH vs. control PA SMC. It is hard to make much of differences in immunoblots and immunohistochemistry because they rely on the affinity of the relative antibodies. The authors should be sure that the siRNAs they use are specific, ie. Do Notch 3 and Notch 2 reduce with Notch 2 siRNA? Does Notch 3 but not Notch 2 decrease with siRNA for Notch 3. Those data should be included. Finally it does appear from the mRNA data in the supplement that the effect of Smad6 induced by TNF α is different, increasing Notch 2 and reducing Notch 3. Thus the TNF α effect may be selective to Notch 3 particularly in the setting of loss of BMPR2. In the mice, knockout of Notch 3 is protective, so there is more to try to reconcile either in the text or in the discussion. It should also be noted that the PSMC come from familial PAH with a BMPR2 mutation so here again the populations are different.

Response 6: *The analysis of NOTCH signalling in the manuscript by Li et al was confined to NOTCH3 and did not assess contributions from other NOTCHs. The authors used an antibody from Santa Cruz for detection of the NOTCH3 ICD (catalog number: sc-7424). We tested this antibody in cell lysates, and despite extensive experience with immunoblotting, we found that this antibody performed very poorly compared with the antibodies we sourced from Cell Signaling Technologies. We confirmed the identity of the bands detected by NOTCH antibodies used in our experiments by siRNA knockdown. These data can be found in Supplementary Fig. 13g.*

One of the central observations in the Li et al. manuscript, as mentioned by the reviewer, was that NOTCH3 knockout mice are protected from hypoxia-induced pulmonary hypertension. We would strongly contest whether this observation supports a specific role for NOTCH3 in the development of pulmonary hypertension, since Notch3 knockout mice demonstrate grossly abnormal arterial maturation in all vascular beds with altered myogenic responses and structural defects (Domenga et al. Genes Dev 2004). In view of this, it is not surprising that a mouse deficient in Notch3 during

development exhibits a deficient response to chronic hypoxic exposure, as well as any other contractile or remodelling stimulus. In addition, the therapeutic benefits of DAPT used in this manuscript could equally well implicate a role for any of the NOTCHs in vascular remodelling. We have reviewed the evidence in the PAH/Notch literature extensively in response to Reviewer 1 (see above) and do not think that our findings are at odds with the general conclusion that cell specific inhibition of Notch may have protective or deleterious effects in PAH depending on the ligand and on the Notch signalling pathway activated. Furthermore, we have highlighted the fact that our observations are related to the disease-relevant scenario of reduced BMPR-II, where non-BMP pathways may be perturbed compared to a normal BMP status. This is particularly pertinent when considering pathways such as NOTCH and src, both of which interact directly with BMP signalling. We stress the importance of BMPR-II deficiency throughout the revised Discussion.

As the immunoblots in Figure 4 would not have been exposed to equal extents, we felt that comparing these blots for the expression levels of NOTCH proteins was not appropriate. To address this question by the reviewer, we conducted Western blots for the NOTCH proteins using lysates from 3 different control and HPAH PASM lines all serum-restricted and lysed in parallel. The Western blots and densitometry graph have been included as Supplementary Fig. 11a and 11b, respectively. We have also added these data into the Results section on page 9. We did not observe any significant differences between the protein levels of NOTCH1, NOTCH2 and NOTCH3 in control and HPAH PASCs. This is consistent with the BMPR2 siRNA data already presented in Supplementary Fig. 14 showing that BMPR2 siRNA does not alter basal expression of NOTCH1, NOTCH2 or NOTCH3.

Comment 7: Page 11: The selective effects of the FYN and YES siRNAs should be documented in view of the results.

Response 7: We thank the author for this suggestion. We confirmed in our original experiments that the siRNAs for SRC, FYN and YES only reduced the mRNA expression of their dedicated targets without affecting the expression of the other two genes. We have incorporated these data as Supplementary Fig. 16d.

Comment 8: Interestingly the TNF overexpression does not result in more severe RVSP or RVH when BMPR2 is heterozygous. On the other hand, in the SUGEN/Hypoxia model there is clear suppression of RVSP and RVH by the TNF soluble receptor but less impact on the muscularization of vessels. The authors should discuss these differences.

Response 8: The reviewer is correct that there is no statistically significant difference in RVSP or RVH between SPC/Tnf^{+/-}:Bmpr2^{+/+} and SP-C/Tnf^{+/-}:Bmpr2^{+/-}, although there is between Bmpr2^{+/+} and SP-C/Tnf^{+/-}:Bmpr2^{+/-} animals. We have changed the text in the results section on page 8 to better reflect these results. This may be because the high levels of TNF α produced by the SP-C/TNF^{+/-} background is causing sufficient reduction in BMPR2 levels to cause the changes observed in RVSP and RVH. However, there is a significant difference in RVSP between Bmpr2^{+/-} and SP-C/Tnf^{+/-}:Bmpr2^{+/-}, demonstrating that TNF α has a much more severe affect on RVSP in animals with a BMPR2 heterozygous background.

The reviewer is correct in that etanercept appeared to have a greater effect on RVSP and RVH than on peripheral lung vascular muscularization. However, small measured changes in lung precapillary vascular muscularization in this severe reversal model can be expected to have profound effects on haemodynamics based on the fact that vascular resistance is inversely proportional to the radius to the fourth power (r^4).

Comment 9: Discussion, Page 13: The discussion with respect to Notch seems to short-change the previous observations, particularly related to the prevention of pulmonary hypertension in the Notch 3 knockout mice. There is no question, however, that the mechanistic studies in this paper are comprehensive and well thought through.

Response 9: *We do not intend to short change the previous observation with respect to Notch and indeed, having further reviewed the literature, provide some further discussion of the field in our revised Discussion. Importantly our studies have been performed in models where BMPR-II expression is critically reduced.*

Reviewer #4 (expert in PAH and BMP receptors).

Remarks to the Author:

The manuscript describes an interesting and complex system involving a switch in TGF β receptor super-family usage in PASMCs from an ALK3 / BMPR-II-driven system to an ACTR-IIA / ALK2 system-driven primarily by TNF α degradation of BMPR-II and the induction in expression of various genes including BMP6 and ACTR-IIA. In the background of compromised BMPR-II expression / function as might be found in PASMCs from patients with heritable forms of Pulmonary Arterial Hypertension (PAH) and genetic mouse models of PAH, this alteration in receptor usage results in a switch from an anti-proliferative response of these cells to a pro-proliferative response following stimulation with BMP6. To add to the complexity, TNF α -mediated modulation in the expression of components of the NOTCH pathway with induction and repression of NOTCH2 and NOTCH3 respectively, directly contributes to the pro-proliferative effects of PASMCs in a Src-kinase family-dependent manner. Although the mechanism described in the manuscript is novel and provides several new therapeutic options for the treatment of heritable PAH, I do have some questions that need to be addressed and suggestions for improvement of the manuscript outlined below before it should be considered for publication.

Comment 1: The authors present data showing that TNF α induces degradation of BMPR-II via an ADAM10/17-mediated mechanism that appears to be operative in PASMCs but not in PAECs. The expression of BMPR-II is clearly suppressed in PAECs in response to TNF α stimulation, however the authors do not provide any explanation as to the mechanism in this cell type. Do the authors have any mechanistic insights into how BMPR-II protein levels are regulated in PAECs via TNF α stimulation that could be included in the manuscript?

Response 1: *We have now investigated the mechanism by which TNF α represses BMPR-II in PAECs, showing that the TNF α -dependent repression of BMPR-II is mediated by p65(ReI α). We have now incorporated these data into Supplementary Fig. 1e.*

Comment 2: The authors have used the SP-C/TNF transgenic mouse model to demonstrate that BMPR-II expression in the whole lung of these animals also appears to be under the control of the TNF α pathway. The data presented by the authors seems to imply that the expression of BMPR-II in whole lung homogenates is almost completely suppressed in this model. This finding seems incongruent with the alveolar epithelial cell-restricted expression of TNF α previously described in this model (Miyazaki et al. 1995 J Clin Invest. 1995 Jul; 96(1):250-9). Immunohistological analysis of BMPR-II and TNF α expression in the lungs of these animals would provide additional spatial information confirming that loss of BMPR-II expression is co-localised to areas of TNF α expression.

Response 2: *The SP-C/TNF transgenic mouse is a good model to study lung specific effects of TNF α . However, although the expression of TNF α is restricted to alveolar epithelium TNF α is a soluble ligand, which on release results in high levels of TNF α in lavage fluid and elevated serum TNF α levels (Fujita 2001 Am J. Physiol. Cell Mol. Physiol. 280:L39-L49.). These data indicate that the pulmonary vasculature will be directly exposed to TNF α and therefore, immunostaining for the cells expressing TNF α would not provide the relevant information relating to the action of TNF α on target cells.*

Comment 3: The legend for Figure 1e needs to have a better description of what stains / proteins the colours in the images refer to. From the images presented, it also appears that α SMA positive cells appear to express TNF α which might suggest that PSMCs isolated from patients with heritable PAH may express higher baseline levels of TNF α in culture. Have the authors determined whether this is the case?

Response 3: *We have edited the figure legend as requested. We have examined RNA samples from control and HPAH PSMC cell lines for the expression of TNF α but did not detect any TNF α mRNA in these cells (CT values all >34 with B-actin Cts = 17-19).*

Comment 4: The authors have cited information showing that the BMPR-II cytoplasmic tail can act as a scaffolding element that can bind to and modulate the activity of a number of kinases including c-Src (Wong et al. Am J Respir Cell Mol Biol. 2005 Nov;33(5):438-46). Can I presume that the generation of the BMPR-II ICD might still be capable of binding to kinases like c-Src and modulating their activity? If so, it would then be important to know how long the BMPR-II ICD remains within the cytoplasm.

Response 4: *This is a very interesting question. We identified during our experiments that the BMPR-II ICD is very labile and can only be detected if the cells are snap-frozen prior to lysis to inhibit this rapid degradation. It would be interesting to establish whether accumulation of this fragment alters intracellular signalling pathways, but the labile nature of this product has rendered further investigation very difficult.*

Comment 5: Previously published information by the authors has shown that PSMCs isolated from patients with PAH lose their growth suppressive responses to BMP ligands including BMP2 and BMP4 (Morrell et al. Circulation. 2001 Aug 14; 104(7):790-5), which is in contrast to the data depicted in Figure 2D. Could the authors provide an explanation for the differences observed in this study compared to the previously published report?

Response 5: *The 2001 Circulation manuscript described a loss of growth inhibition to BMP ligands in PSMCs from patients with idiopathic and heritable PAH. This loss of growth inhibition was not complete, was dependent on the concentration of BMP, and was more evident in cells from BMPR2 mutation carriers than PAH patients without mutations. Figure 2D also shows that PSMCs from BMPR2 mutation carriers are less susceptible to growth suppression compared with control cells. Thus the data are not inconsistent with our previous work. The experiments conducted in the present manuscript were in 5% serum rather than 0.1% serum used in the 2001 paper.*

Comment 6: There are issues with Figure 4 and supplementary Figures 11 and 14:
a) Presumably Figure 4a, 4b and Supplementary Figure 11a, 14d depict NOTCH1, NOTCH2, and NOTCH3 ICD and not full-length protein as indicated in the main body of the manuscript. The Figure legends need to be clarified that what is being depicted are indeed the ICD's and not full-length NOTCH proteins.

Response 6a: *We thank the reviewer for identifying these discrepancies. As addressed by reviewer 2, the blots represent the S1 cleaved products of NOTCH1, NOTCH2 and NOTCH 3 rather than the ICDs. We have edited all previous references to the ICD to reflect that we are examining the NOTCH cleaved/transmembrane intracellular (NTM) regions in the Results section (page 9). We have now adjusted the figure legends for Figure 4 and Supplement Figures 11, 13 and 14 to reflect this.*

b) On page 9 of the manuscript the authors state that in Figure 4a and Supplementary Figure 14c 'BMPR2 silencing also promoted the TNF α -dependent reduction of NOTCH3-ICD generation and NOTCH3 transcription'. On inspection of Figure 4a, TNF α does induce a reduction in NOTCH3 ICD levels; however, BMPR2 siRNA does not enhance the effect of TNF α on NOTCH3 ICD levels compared to the siCP or DH1 control. Similarly, in Supplementary Figure 14c, TNF α does reduce the transcription of NOTCH3 mRNA; however, this effect is not enhanced by BMPR2 siRNA compared to the siCP or DH1 controls.

Response 6b: *The reviewer is correct that the reduction of Notch3 after siBMPR2 knockdown was observed in the co-treatment of TNF α and BMP6 we have adjusted the manuscript to reflect this Page 10).*

c) Also on page 9 of the manuscript the authors state in Figure 4b 'In BMPR2 heterozygous HPAH PSMCs treated with TNF α , siACVR2A reduced NOTCH2-ICD generation and abrogated NOTCH3-ICD'. On inspection of Figure 4b, treatment of HPAH PSMCs with siACVR2A seems to be without effect on TNF α -mediated NOTCH2-ICD generation.

Response 6c: *Figure 4b shows that the increase in NOTCH2 in HPAH PSMCs treated with TNF α and BMP6 is reduced by siACVR2A, but we agree with the reviewer that the impact of TNF α alone is minimal. We have altered this sentence on Page 10 to describe more accurately what was observed.*

d) In Figure 4b, treatment of HPAH PSMCs stimulated with 0.1% serum with DH1 alone seems to have had a significant suppressive effect on NOTCH1 protein expression. This does not appear to be due to a loading issue as the authors state that the gels were re-probed for α -tubulin which appears

to be uniformly expressed in all lanes on the gel. As this gel is representative of the other replicates, is the effect of DH1 on NOTCH1 expression also observed in the other replicates?

Response 6d: *The reviewer is correct to point out that there is an apparent reduction in the DH1 lane in Figure 4b. However, we would interpret this to be due to an increase in the siCP band for NOTCH1 in this blot. This cannot be explained by unequal loading as the reviewer points out. Indeed, we have now repeated this blot a further 4 times in different mutant cell lines and have observed frustratingly similar effects in all of these. We cannot explain this observation, especially since we do not observe this effect in control cells. Nevertheless, this does not detract from our general conclusion that TNF α has modest effect on NOTCH1, compared with NOTCH 2 and 3, which is further confirmed at the mRNA level in Supplementary Figures 11 and 14. Moreover, we do not implicate NOTCH1 further in the signalling pathways downstream of the TNF α effect. In light of this we feel it is appropriate to remove the NOTCH1 blots from the main paper (Figure 4A and B). We thought it would be useful to retain the data in the supplementary section to indicate that we have examined protein levels of his NOTCH, whilst acknowledging the technical difficulty with the siCP in mutant cells. Thus, we have added these data to Supplementary Figure 14e-f and make mention of this issue in the Figure legend.*

Comment 7: The authors have utilised the γ -secretase inhibitor, DAPT in experiments to support the notion that NOTCH ICD generation is involved in the TNF α and BMP6-induced proliferative responses of PSMCs. The concentration of DAPT used in this study was 5mM. Are the authors confident that other proteases important in their proposed mechanism are unaffected by this concentration of DAPT? How does 5mM DAPT affect TNF α -mediated generation of BMPR-II ICD?

Response 7: *We thank the reviewer for drawing our attention to this error. This may have been due to an oversight in converting the “m” to a “ μ ” in Symbol font. We can confirm that the final concentration of DAPT used for the experiment was 5 μ M and have corrected the manuscript accordingly.*

Comment 8: In Figure 4e, the authors state that expression of NOTCH2 in the medial layer of the pulmonary arteriolar lesion depicted in the lung section taken from an HPAH patient is enhanced compared to control vessels. The intensity of the staining looks somewhat similar between the normal vessel and HPAH vessel in the sections depicted.

Response 8: *As I am sure the reviewer can appreciate that ascertaining the expression level of proteins from immunohistological staining can be challenging. We agree with the reviewer that the intensity of Notch2 appears to be similar in both sections but many more cells in the medial layer of the concentric lesions possessed Notch2 staining. We have therefore adjusted our description in the manuscript to reflect this (page 10).*

Comment 9: In Supplementary Figure 17b, Etanercept did not change the % wall thickness for all vessels yet in Figure 6c there appears to be a modest effect of Etanercept on the % of muscularized vessels with Etanercept causing a small increase in non-muscularized vessels compared to the S/H control. Are these endpoints essentially measuring the same thing i.e. pulmonary vascular wall remodelling and, if so, is the data inconsistent?

Response 9: *The reviewer is correct to point out that the dose of etanercept used at the time point studied is not significantly changing the percentage wall thickness, yet is reducing the percentage of muscularised vessels in precapillary small arterioles. The wall thickness is assessed in larger vessels associated with small (terminal bronchiole) airways, whereas the determination of percentage of muscularised vessels is established by examining the smaller alveolar duct-associated vessels in the very periphery of the pulmonary arterial bed. Therefore, these measurements are not measuring the same thing. The process of pulmonary vascular remodelling is thought to initiate in the most distal vascular bed (measured by distal muscularisation), whereas the wall thickness in larger arteries (wall thickness) probably reflects the increased intravascular pressure. When using a therapeutic agent to reverse established disease, (especially one that does not work by vasodilatation) one would expect the initial reversal to occur in the very distal vascular bed and the reduction in wall thickness of larger arteries to take longer to resolve as the pressures come down. The sugen-hypoxia rat model employed in our manuscript sets a very high bar for reversal of PAH given the severity of disease and the relatively short period of therapeutic intervention. Nevertheless, we observed significant reversal of muscularisation in the distal pulmonary circulation, not forgetting small measured changes in lung precapillary vascular muscularization in this severe reversal model can be expected to have profound effects on haemodynamics based on the fact that vascular resistance is inversely proportional to the radius to the fourth power (r^4).*

We have clarified the methods section to reflect these different methods of assessment and have changed the following sentence in the results (Page 12) to refer to this observation.

“Etanercept reversed the progression of PAH, reducing RVSP, right ventricular hypertrophy and muscularization of small alveolar duct-associated arterioles (Fig 6a-c), but not the wall thickness of the larger arterioles associated with terminal bronchioles (Supplementary Table 17b), without altering left ventricular function (Supplementary Table 2).”

Comment 10: In the body of the text, the authors state repeatedly that the transient induction of Smad1/5 responses by BMP6 / ACTR-IIA / ALK2 signalling is unlikely to promote heightened PASMCM proliferation. There are no data shown in the manuscript or supplementary materials that indicate whether the Smad1/5 responses mediated by BMP6 / ACTR-IIA / ALK2 are transient or otherwise. If the authors cannot qualify this statement with data or a citation, then the authors should desist from describing the responses as transient.

Response 10: *To confirm the transient nature of this Smad response we have included new experiments demonstrating that BMPR-II knockdown leads to increased p-Smad1/5 activity in response to BMP6 (Supplementary Figure 7b and 7c) but that this response is very limited in duration (<4 hours) and describe this in the Results section (Page 7) In contrast, the sustained signalling pathway identified by our studies as responsible for hyper-proliferation of PASMCMs is robustly shown to be aberrant Notch signalling downstream of ALK2/ActRIIa.*

REVIEWERS' COMMENTS:

Reviewer #1 (Remarks to the Author):

The authors have provided a robust response to comments and questions.

Reviewer 1, Comment 4: I think the complexity of the field prevents a satisfactory conclusion to this comment. In other words the data in their paper are internally consistent but at odds with other published data and we might have to leave it there, recognising that the findings might be peculiar to the conditions of the experiment. This should temper conclusive statements.

Comment 3: It is not easy to understand how the mechanism applies solely to the pulmonary circulation given that the inflammation (and elevated circulating TNF α) is a systemic effect in pulmonary hypertension i.e. not confined to the pulmonary vascular bed. This cannot be brushed aside because there may be important biology here but this study does not set out to address this and can only acknowledge it as a limitation to their conclusions.

Reviewer #2 (Remarks to the Author):

The authors have addressed my concerns in a positive and satisfactory manner. The only very minor issue that remains to be addressed is that the molecular weight markers next to the bands representing ADAM10 and ADAM17 in the blots shown in figure 1h and in supplementary figure 5c don't match (ADAM17: ~130 kD in 1h versus ~90 kD in 5c; ADAM10: ~100 kD in 1h versus ~75 kD in 5c). Please address this discrepancy and make sure that the correct molecular weight markers are indicated.

Reviewer #3 (Remarks to the Author):

the authors have addressed the comments and suggestions of my review.

Reviewer #4 (Remarks to the Author):

I thank the authors for addressing most of my queries.

I have the following outstanding points:

- The authors cite the paper from Fujita et al. 2001 stating that elevated levels of TNF α were detected in the serum indicating that TNF α was freely diffusible thus dispelling the requirement to conduct any immunohistochemistry (IHC). On reading the paper from Fujita and colleagues, it seems that there was a non-significant trend to increased TNF α in the serum observed, whereas the levels of TNF α in lavage fluid was significantly elevated as would be expected if the cytokine was produced and liberated from the epithelial surface. To my mind, conducting this IHC analysis would add value to the manuscript.
- I thank the authors for correcting the "m" to " μ " for the concentration of DAPT used. In relation to my original question, what impact did 5 μ M DAPT have on the TNF α -mediated generation of the BMPRII ICD?

REVIEWERS' COMMENTS:

Reviewer #1 (Remarks to the Author):

The authors have provided a robust response to comments and questions.

Reviewer 1, Comment 4: I think the complexity of the field prevents a satisfactory conclusion to this comment. In other words the data in their paper are internally consistent but at odds with other published data and we might have to leave it there, recognising that the findings might be peculiar to the conditions of the experiment. This should temper conclusive statements.

Response: *We appreciate this comment by the reviewer. We have endeavoured to stress that our data focuses on the interaction of TNF α and BMPR2.*

We have included the following sentence in the discussion:

“We acknowledge that our data contradicts some aspects of previous reports regarding the contribution of NOTCH signalling in PAH, but our studies primarily focus on the combined impact of TNF α and a background of BMPR2 haploinsufficiency, which is the context relevant to the majority of human heritable cases.”

Comment 3: It is not easy to understand how the mechanism applies solely to the pulmonary circulation given that the inflammation (and elevated circulating TNF α) is a systemic effect in pulmonary hypertension i.e. not confined to the pulmonary vascular bed. This cannot be brushed aside because there may be important biology here but this study does not set out to address this and can only acknowledge it as a limitation to their conclusions.

Response: *Although we have previously reported raised systemic levels of circulating TNF α are found in PAH patients (Soon et. al. 2010 Circulation), the TNF α levels are 7.92pg/ml in controls, 10.45pg/ml in IPAH and 9.85 pg/ml in HPAH, representing low levels of activity. In this current manuscript, we show appreciable local expression of TNF α in the media of pulmonary arteries from PAH patients and we consider that the local pulmonary vascular levels will have a far greater effect on the smooth muscle cells across the entire thickness of the vessel wall. Conversely, the slightly elevated systemic levels are likely to exert lesser effects in the systemic circulation. Moreover, this question is a bit like asking why BMPR2 mutations only affect the lung circulation? Although we don't know the full answer to this question, it is clear that suppression of BMPR2 signalling, by TNF α or by any other means, has an effect that is specific to the lung circulation.*

We have now added the following text (new text underlined) into the discussion to reflect this:

“Inflammatory cytokines are associated with the pathogenesis of PAH^{8,9,12} and we demonstrate that local TNF α expression is present in the medial layers of pulmonary arteries from IPAH and HPAH patients, but not in control tissues. Given that we previously reported elevated systemic circulating TNF α levels of 10.45pg/ml in IPAH patients and 9.85 pg/ml in HPAH patients compared to 7.92pg/ml in controls¹², we suggest that local lung expression is likely to generate appreciably higher TNF α levels and have a more restricted effect on the pulmonary circulation than small elevation in these relatively low systemic levels.”

Reviewer #2 (Remarks to the Author):

The authors have addressed my concerns in a positive and satisfactory manner. The only very minor issue that remains to be addressed is that the molecular weight markers next to the bands representing ADAM10 and ADAM17 in the blots shown in figure 1h and in supplementary figure 5c

don't match (ADAM17: ~130 kD in 1h versus ~90 kD in 5c; ADAM10: ~100 kD in 1h versus ~75 kD in 5c). Please address this discrepancy and make sure that the correct molecular weight markers are indicated.

Response: *We can confirm that the markers for the respective ADAM10 and ADAM17 differ between the blots in Figure 1h and Supplement Figure 5c. The differences between the data in these figures are due to a difference in the molecular weight of the proteins detected in human PSMCs (Figure 1h) compared to mouse lung (Supplement Figure 5c), with the human isoforms being larger. We have confirmed the identity of both proteins in PSMCs using specific siRNAs. It may be pertinent that human ADAM10 and ADAM17 are both subject to splice variation.*

Reviewer #3 (Remarks to the Author):

the authors have addressed the comments and suggestions of my review.

We thank the reviewer for their comments.

Reviewer #4 (Remarks to the Author):

I thank the authors for addressing most of my queries.

I have the following outstanding points:

- The authors cite the paper from Fujita et al. 2001 stating that elevated levels of TNF α were detected in the serum indicating that TNF α was freely diffusible thus dispelling the requirement to conduct any immunohistochemistry (IHC). On reading the paper from Fujita and colleagues, it seems that there was a non-significant trend to increased TNF α in the serum observed, whereas the levels of TNF α in lavage fluid was significantly elevated as would be expected if the cytokine was produced and liberated from the epithelial surface. To my mind, conducting this IHC analysis would add value to the manuscript.

Response: *We concur that the levels of TNF α in the serum of SPC/Tnf mice, reported as being 35pg/ml compared to 17.5pg/ml in WT littermates, is lower than the 60-100ng/ml detected in the BALF fluid. As the lung comprises an enormous area of 1-2 cell thick alveoli, it is highly likely that TNF α , which promotes permeability, will be readily able to act on the pulmonary capillary network to reduce BMPR-II expression. Furthermore, in situ hybridisation analysis in the original manuscript demonstrated that the SPC/Tnf mice express the Tnf transgene mRNA across the entire lung (Miyazaki et al. 1995 J.Clin.Invest. 1995 Jul; 96(1):250-9).*

Moreover, this SPC/Tnf mouse used in our paper has been studied by several other groups and the summation of this work is that this mouse exhibits significant elevations of serum TNF α as exemplified in the following paper:

Tang K, Murano G, Wagner H, Nogueira L, Wagner PD, Tang A, Dalton ND, Gu Y, Peterson KL and Breen EC. J.Appl.Physiol. 2013 May;114(9):1340-50. Impaired exercise capacity and skeletal muscle function in a mouse model of pulmonary inflammation. doi:10.1152/jappphysiol.00607.2012.

In addition, IHC for TNF α has been undertaken before in the lungs of this mouse. These studies have shown that TNF α is observed both within epithelial cells and the surrounding tissues, as would be expected from a diffusible cytokine:

Eurlings IM, Dentener MA, Mercken EM, de Cabo R, Bracke KR, Vernooij JH, Wouters EF and Reynaert NL. Am.J.Physiol.Lung.Cell.Mol.Physiol. 2014 Oct 1;307(7):L557-65. A comparative study of matrix remodeling in chronic models for COPD; mechanistic insights into the role of TNF- α . doi:10.1152/ajplung.00116.2014.

On the basis of these reports it would seem unnecessary for us to undertake the IHC suggested by the reviewer.

- I thank the authors for correcting the “m” to “ μ ” for the concentration of DAPT used. In relation to my original question, what impact did 5 μ M DAPT have on the TNF α -mediated generation of the BMPR-II ICD?

Response: Although the reviewer raises this question we are not clear how performing this experiment would add anything to our manuscript. DAPT is a gamma secretase inhibitor and there is no scientific reason, based on any published evidence, to suspect that it will impact on the shedding of BMPR-II. Even if it did, that observation would not change any of the conclusions of our paper.